# Real-time outage management in active distribution networks using reinforcement learning over graphs

Roshni Anna Jacob[1,7], Steve Paul [2,7], Souma Chowdhury [2,3,8] ✉,
Yulia R. Gel[4,5] & Jie Zhang [1,6,8] ✉

Self-healing smart grids are characterized by fast-acting, intelligent control mechanisms that minimize power disruptions during outages. The corrective actions adopted during outages in power distribution networks include reconfiguration through switching control and emergency load shedding. The conventional decision-making models for outage mitigation are, however, not suitable for smart grids due to their slow response and computational inefficiency. Here, we present a graph reinforcement learning model for outage management in the distribution network to enhance its resilience. The distinctive characteristic of our approach is that it explicitly accounts for the underlying network topology and its variations with switching control, while also capturing the complex interdependencies between state variables (along nodes and edges) by modeling the task as a graph learning problem. Our model learns the optimal control policy for power restoration using a Capsule-based graph neural network. We validate our model on three test networks, namely the 13, 34, and 123-bus modified IEEE networks where it is shown to achieve near-optimal, real-time performance. The resilience improvement of our model in terms of loss of energy is 607.45 kWs and 596.52 kWs for 13 and 34 buses, respectively. Our model also demonstrates generalizability across a broad range of outage scenarios.

Resilience enhancement of power distribution networks (DNs) has been gaining considerable recognition in recent years, which has been often overlooked before due to the perception of DNs as merely a link between the transmission networks and consumers. A key factor for this shift is the realization that 90% of customer disruptions during extreme events can be attributed to the failure of components within the distribution network itself[1]. Additionally, the increasing presence of distributed energy resources (DERs) and the resulting decentralization of power generation have spurred the notion of DNs as autonomous entities that can operate independently from the main grid[2]. Consequently, the DN is now considered capable of retaining its functionality even during the loss of connectivity to the transmission network.

Concurrently, modernization of the power grid and the shift toward smart grids have been driving the deployment of intelligent and automated technologies in the DN[3]. The distribution automation

[1]Department of Electrical and Computer Engineering, The University of Texas at Dallas, Richardson, TX 75080, USA. [2]Department of Mechanical and Aerospace Engineering, University at Buffalo, Buffalo, NY 14260, USA. [3]Department of Computer Science and Engineering, University at Buffalo, Buffalo, NY 14260, USA. [4]National Science Foundation, Alexandria, VA 22314, USA. [5]Department of Mathematical Sciences, The University of Texas at Dallas, Richardson, TX 75080, USA. [6]Department of Mechanical Engineering, The University of Texas at Dallas, Richardson, TX 75080, USA. [7]These authors contributed equally: Roshni Anna Jacob, Steve Paul. [8]These authors jointly supervised this work: Souma Chowdhury, Jie Zhang. ✉e-mail: soumacho@buffalo.edu; jiezhang@utdallas.edu

has been implemented through the deployment of line monitors, fault indicators, remote-controlled switches, and reclosers in the DN[4]. An important characteristic of the smart grid is its self-healing capability, which includes implementing intelligent control actions through automation to minimize power disruptions, thus enabling the recovery of network operations during outages in real time[5]. Therefore, the key requirements of a self-healing tool include autonomy, quick response, and online adaptability, which are indeed the salient features of our model discussed in this paper.

The transformation of the grid to a smart grid is driven by a bottom-up approach[6] with distribution feeders interacting at the transmission level. This paper specifically explores the intricacies of the lower-level component of the smart grid - the distribution network. The smart grid typically operates as an independent entity governed by an independent system operator (ISO). Intergrid operations are challenging due to differing protocols, communication systems, and regulatory jurisdictions among independent system operators (ISOs). Additionally exploring new frontiers in smart grid operation is constrained by the ongoing development of communication infrastructure standardization and interoperability. The operation and control of the distribution networks within the smart grid are mostly autonomous with its aggregated impact visible on the transmission level[7]. However, inter-grid operations are seldom employed during extreme events, driven by concerns about potential cascading failures between independent entities.

In the face of power disruptions caused by extreme weather events or cyber-physical attacks, a self-healing DN warrants the automatic detection of faulty components, their isolation, and system restoration (fully or partially) using intelligent control algorithms. This process is referred to as FLISR, which stands for fault location, isolation, and service restoration[5], and is addressed using task-specific techniques. Restoration or the recovery of DN operation can be achieved using different control actions, such as network reconfiguration, load management, DER control, energy storage control, and reactive power resource control. The preliminary control action often adopted in such circumstances is reconfiguration (or switching control), followed by load shedding[8,9]. Distribution network reconfiguration (DNR) by controlling the status of the network switches is a commonly used strategy to control DN operation for varying objectives such as loss minimization, reliability enhancement, load balancing, increasing penetration of renewable resources, improvement of voltage profile, and service restoration[10–12]. The purpose of feeder reconfiguration is two-fold: (1) to quickly and efficiently reroute power from the functional part of the DN to the isolated section[13,14], and (2) to form intentional islands around the grid-forming DERs when there exists no connectivity to the main grid[2,15]. In the existing body of knowledge, these two reconfiguration strategies have been addressed separately and have been largely considered as two distinct domains. However, a comprehensive restoration strategy suitable for various outage scenarios must efficiently utilize both grid-forming and grid-feeding DERs and consider all possible reconfiguration (or switching) options[16]. Hence, in our framework, we consider both DN characterizations through switching by including simultaneously grid-connected and off-grid modes of operation. Additionally, DNR alone may not be sufficient as a restorative action during catastrophic events, as the network remains vulnerable to voltage collapse and system blackouts[9,17]. Therefore, load shedding becomes necessary as an emergency control mechanism[18] to minimize voltage violations in the DN.

Furthermore, power distribution networks are typically unbalanced and radial in nature, with a unidirectional power flow from the substation to the consumers. Besides the non-linearity in power flow, the optimization of modern-day DN operation has also been made challenging by the integration of DERs[19]. The DN restoration is an NP-hard, non-linear combinatorial optimization problem that aims to maximize energy supply while considering network connectivity and operational constraints[20]. Various methods have been used in the literature to solve the traditional reconfiguration problem, falling into heuristic[21,22], meta-heuristic[23–25], and mixed-integer programming[26–28] techniques. In line with the increasing penetration of DERs, researchers have also explored islanding strategies using mixed-integer programming models to expand the zone of DER operation[2,29]. Load management during outages has also been previously investigated as an emergency control strategy[16,30]. Despite these efforts, a solution incorporating both the grid-connected and islanding (off-grid) reconfiguration schemes for outage management is limited in literature, and presents a complex and challenging problem to solve. The multitude of restorative options depends on the number of controllable devices (switches, loads) and the operational modes of DERs. Although the proliferation of remote-controlled elements in the DNs widens the horizons of automated network control, it also increases the complexity of the underlying non-linear combinatorial optimization problem[31]. The commonly used mixed-integer non-linear programming (MINLP) methodologies for restoration problems face issues of scalability, computational tractability, and real-time decision-making capability[16]. Apart from these, the existing linear programming approximation models in the literature are not designed to address restoration in three-phase unbalanced DNs with sectionalizing, tie switches, and various types of DERs (grid-forming and grid-feeding). Heuristic and meta-heuristic techniques, although explored, tend to be computationally expensive and time-consuming. Moreover, traditional methods heavily rely on a comprehensive description of the DN model and network parameters, making them model-dependent. Considering the uncertainty in network conditions during outages, it is desirable to develop a model capable of adapting to varying circumstances and is deployable online. Here, we present a model based on reinforcement learning to provide online decision support during outages.

Reinforcement learning (RL) methods have been increasingly adopted in recent years for power system applications that require autonomous control[32]. This is because RL methods are quite effective in solving high-dimensional, combinatorial, stochastic optimization problems, besides providing fast-acting control. The latter is imperative to rapid responsiveness during outages, otherwise not possible with conventional optimization-based decision support. Deep RL is being increasingly employed for voltage control in active DNs in recent literature. In ref. 33, the DER inverters and static VAR compensators were controlled to achieve the desired voltage levels in the network using a combination of graph-based network representation learning, surrogate model of power flow, and soft actor-critic algorithm. In ref. 34, the distributed energy storage devices have been treated as agents, and a multi-agent deep RL was utilized for voltage regulation with the capability to respond to topology changes as well. In another study[35], multi-agent deep RL was applied to perform optimal scheduling of various DERs, energy storage systems, and flexible loads within the network. In this context, the inverters associated with DERs and energy storage can be considered as individual agents. The role of such devices in voltage regulation aligns with the distributed nature of their control mechanism. Conversely, outage management using reconfiguration and load control relies on wide-area measurements at the control center to facilitate switching operations. Particularly with regards to reconfiguration, RL-based models[36,37] have been developed to perform dynamic DNR during normal operation for loss minimization and voltage improvement. These methods specifically used deep Q-learning with neural networks and trained the off-policy RL network using a historical network operation dataset. The exploration problem that may arise in these models has been addressed by a Noisy-Net Q-learning model[38] developed to perform DNR for similar objectives. Another approach[39], utilized a batch-constrained soft actor-critic algorithm to learn the control policy for loss minimization during

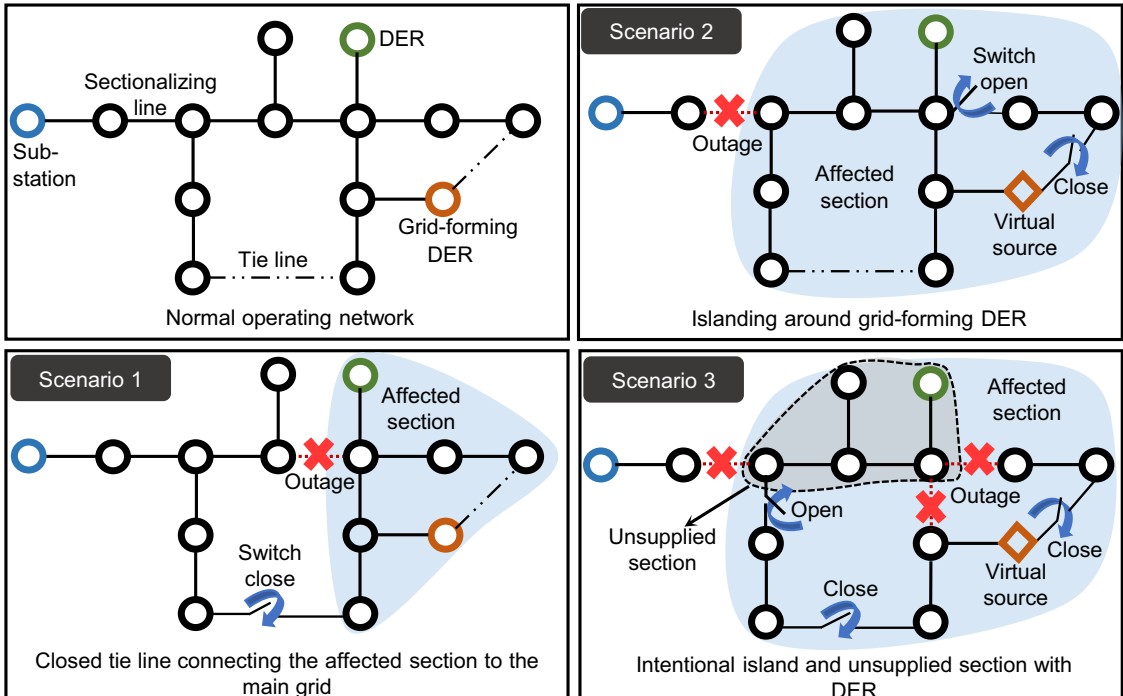

**Fig. 1 | Schematic of an example network with distributed energy resources (DERs) both with and without grid-forming ability, and sectionalizing/tie switches.** The solid lines represent sectionalizing lines, while the dashed lines indicate tie lines. The substation is marked in blue, the grid-forming DER in orange, and the grid-feeding DER in green circles. Outages are denoted as red crosses in the figure. Outage scenario 1 isolates a small portion of the network, which can be restored by closing the tie line connecting the affected section (shaded in blue) to the main grid (substation). In the subsequently connected network formed after reconfiguration, the substation serves as the source bus. In scenario 2, a larger part of the network is disconnected without any means to reroute power from the substation. By selecting suitable switching actions, an intentional island is created around the grid-forming DER in scenario 2. The grid-forming DER serves as the virtual source (marked in orange) in the reconfigured network. In scenario 3, with extensive outage, although an intentional island may be formed around the grid-forming DER, a section of the network remains unsupplied despite the presence of a DER (not grid-forming).

normal DN operation. As opposed to the DNR during normal operation considered in these studies, extreme operating conditions are more challenging considering the high-impact, low-probability occurrence of such events. Therefore, availing historical datasets for network operation may also not be possible as in previous studies. Although researchers have explored using RL models for DNR[40,41] to improve network resilience, such works do not consider the feasibility of network operation based on voltage monitoring and DER operational modes during reconfiguration. Additionally, in methods based on the Q-learning approach, the policy network determines the optimal/near-optimal configuration or the spanning forest, rather than individually controlling each switch. This approach would require enumerating all feasible configurations to define a Q-probability matrix, which is impractical due to the exponential increase in state and action space with network size, possible outage scenarios, and the number of devices. Since these methods are not scalable and require significant storage and computational capabilities for exploration, policy gradient methods are more suitable for learning in outage conditions[39]. We, therefore, employ the proximal policy optimization (PPO), which is a policy gradient method for learning DN outage management in DN. In another work[42], a deep Q-learning-based RL approach was employed to dynamically form microgrids in response to outages. However, this method necessitates the compilation of all radial feasible structures before the learning process and does not encompass both forms of reconfiguration. Similarly, ref. 43 utilized a Q-learning-based strategy for reconfiguration and load shedding. Lastly, in addition to load and switch control, deep RL could also be used for optimal dispatch of DERs in islanded mode as demonstrated in ref. 44.

In this model, our idea is based on the intrinsic graph representation of power distribution networks. The DN is viewed as a graph where nodes are the buses (i.e., substation, load, or DERs) and edges are the lines or transformers. The state variables of the DN, including demand/generation estimates and voltage/current measurements, can be considered as data superimposed on a graph. The state variables exhibit complex interdependencies, necessitating the extraction of meaningful representations that accurately capture the structure of the DN connectivity. Moreover, outage management, particularly reconfiguration, involves altering the DN connectivity by switching on/off network lines (binary actions) and hence, requires consideration of the underlying combinatorial network structure. Therefore, we present Graph RL (GRL) approach for simultaneous real-time control of network topology and loads, ensuring sustained network operations during failures that are caused by extreme events. GRL uses a graph neural network or GNN as a policy model (as is the case here) and/or "value" model, as it allows more effective capturing of the combinatorial nature of network-based state information (involving both binary and continuous variables). This advantage is demonstrated in our case studies through comparison with baseline RL-based solutions that use a standard multi-layered perceptron (MLP) based policy model.

Specifically, we use a Graph Capsule (GCAPS) neural network to learn optimal control policies in power network resilience problems. Compared to other GNNs such as Graph Convolutional Networks (GCN), the capsule-based GNN has been shown by refs. 45–47 to better capture the structural information of a graph (the DN in this work) as a graph embedding, where the individual intermediate features of the state are represented as a vector (in GCAPS) as compared to that of a scalar for example in GCN and Graph Attention Networks (GAT), thus giving an enhanced state representation. This enhanced state representation helps in computing better actions compared to other simple feature abstraction networks such as Multi-Layered Perceptron (MLP).

Experimental validation of our trained GCAPS-based model on test networks demonstrates the generalizability and real-time control capability with near-optimal performance which is desirable in a self-healing tool for DNs.

# Results

## Reconfiguration and load shedding as emergency response

During extreme events in the DN, the occurrence of outages due to component failures can be addressed by a combination of control actions, including reconfiguration and load shedding. We assume that real-time outage detection and protection system responsible for detecting, locating, and isolating faulty components is a preliminary step to the work discussed in this paper.

Line switches in the DN are typically divided into two categories: switches associated with normally-closed sectionalizing lines and those with normally-open tie lines. During emergency conditions, when component failures disrupt the power supply to the network loads, reconfiguring the DN through control actions on these switches can help maintain network functionality. The objective in such situations is to maximize (or minimize) the energy supplied (or loss of energy) to the loads, despite the network failure, while ensuring operational stability. The optimal switching control depends on factors such as the network state (voltage, branch flow, etc.), network operational limits, and the location and extent of the outage in the network.

Besides this, the presence of DERs, particularly grid-forming DERs, plays a pivotal role in providing uninterrupted supply to loads following outages. In the off-grid mode, the formation of a self-sustained entity comprising loads and DERs is only possible with the assistance of grid-forming DERs. These grid-forming DERs generate the reference voltage and frequency for the isolated network section while grid-feeding DERs follow this reference and inject active/reactive power into the grid[48]. While the detailed modeling of these DERs is beyond the scope of this work, they are represented as voltage sources when operating in the grid-forming mode, and this characterization is incorporated in the DN model within the environment.

Reconfiguration is often used as an umbrella term for any change in normal operating network topology using switching control. On the other hand, intentional islanding has long been recognized as a resilience enhancement technique and is a subset of the reconfiguration problem. In scenarios where the outage is extensive and the availability of tie switches is limited, intentional islanding around grid-forming DERs may be adopted to ensure a continuous power supply. Figure 1 illustrates the different switching actions that may be employed based on the extent of the outage. Different outage scenarios are portrayed in Fig. 1 with mitigation strategies representing the possible solutions we considered while designing the environment.

Network topology control through switching actions alone cannot guarantee the operational feasibility of the energized sections in the network. Therefore, to ensure sustainable network operation, emergency load shedding is also considered to maintain network voltage within safe operational limits. The loads are modeled as equivalent load at the distribution transformer in the primary distribution system and can be disconnected from the network through switching actions.

## DN representation as a graph

Outage management in DN using switching control can be largely viewed as a task of learning the associated network topology, which is our motivation to reformulate the problem in graph-theoretic terms. Consequently, we represent the DN as a graph $\mathcal{G} = (\mathbf{N}, \mathbf{E})$, with an $\mathbf{N}$ set of nodes interconnected by an $\mathbf{E}$ set of edges. The nodes in the graph represent the buses in the DN, including the substation, load, DER, and zero-power injection buses. The edges represent the distribution lines and inline transformers. These lines (edges) consist of both switchable

(sectionalizing and tie) and non-switchable lines. The node variables comprise both forecasted or estimated variables and measured variables. These variables include the estimated or forecasted values for active power demand (or generation), reactive power demand (or generation), and the three-phase voltage measured at each bus. The edge variable considered is the measured power flow through the branches. To obtain these measured signals, we utilize a power flow simulator in our synthetic approach.

Network reconfiguration in the graph domain essentially involves determining the status (open or closed) of the switchable edges in the DN. Emergency load shedding at the primary DN level is indicated using a binary variable associated with the nodes representing switchable loads.

## A Markov decision process over graphs

The emergency response during outages in the DN is formulated as a Markov Decision Process (MDP) in the graph domain, denoted as $\mathcal{M} = (\mathcal{S}, \mathcal{A}, \mathcal{P}_{tr}, \mathcal{R})$. The tuple denotes the state, action, transition probability, and reward (in the respective order), which are defined as follows:

(1) State ($\mathcal{S}$): the state is composed of relevant observations from the DN that represent the current operating condition of the network. It includes node variables, edge variables, network topology, and other system variables, denoted as $\mathcal{S} = [P_d^N, Q_d^N, P_g^N, Q_g^N, V^N, V_{viol}, I^E, \mathcal{T}, E_{supp}, \mathcal{O}, \mu]$. Here, $P_d^N, Q_d^N$ represents the estimated or forecasted active and reactive power demand at the nodes, while $P_g^N, Q_g^N$ corresponds to the active and reactive power generation at the nodes. The three-phase voltage measured at the buses (graph nodes) is represented as $V^N$, and $V_{viol}$ indicates the voltage violation in the network. The edge variable includes the power flow through the network branches, denoted as $I^E$. The operating topology of the network is $\mathcal{T}$, and the total energy supplied in the network is represented by $E_{supp}$. The variable $\mathcal{O}$ in the state encapsulates the outage scenario, i.e., the multi-line failures in the network, including switch outages. The inoperability of the outage switches is addressed by using a masking mechanism that suppresses the corresponding switching action, represented by the state variable $\mu$.

(2) Action ($\mathcal{A}$): the control actions for emergency response include switching and load shedding. Therefore, the action space is represented as $\mathcal{A} = [\delta_1^{sw}, \delta_2^{sw}, ..., \delta_{N_S}^{sw}, \delta_1^{ld}, \delta_2^{ld}, ... \delta_{N_L}^{ld}]$. Here $N_S$ represent the number of switchable lines, which includes both the sectionalizing and tie lines. The number of switchable loads in the network is denoted as $N_L$. Line switching is represented by a binary variable $\delta^{sw}$ where 0 and 1 represent the opening and closing of the switch, respectively. The status of the loads is also represented by a binary variable $\delta^{ld}$, where load served and load shed respectively corresponds to 1 and 0.

(3) Transition probability ($\mathcal{P}_{tr}$): the transition probability captures the dynamic nature of the network with emergency response, denoted as $\mathcal{P}(s'_{t+1}|s_t, a_t)$. This represents the transition from network state $s$ at time step $t$ to state $s'$ at step $t+1$ given that action $a$ is implemented at time step $t$. The transition probability is learned by the agent from its interactions with the environment.

(4) Reward ($\mathcal{R}$): the reward guides the GRL algorithm to take optimal control actions for mitigating outages in the DN, which is formulated as follows:

$$r(s, a) = \begin{cases} E_{supp} - V_{viol}, & \text{if } C_{viol} = 0, \\ 0, & \text{otherwise.} \end{cases} \tag{1}$$

The reward reflects the goal of improving resilience in the DN by maximizing the energy supplied $E_{supp}$ while minimizing violations of voltage constraints. To account for the network being ill-conditioned with specific outage conditions and switching actions, a term $C_{viol}$ is

introduced into the reward. The DN, subject to topology changes due to outages and switching actions, may consist of multiple independent sections (network components), each housing various active components (transformers, regulators, generators, loads, etc.) with corresponding state variables. In some scenarios, the isolation of these components from a robust slack (substation) renders the network ill-conditioned, resulting in challenges in achieving nodal power balance within a preset tolerance of mismatch. This lack of balance in certain sections of the DN leads to non-convergence of power flow, identifiable through flags in the solver. This issue is attributed a zero value with the actual impact of switching on the network state being indeterminate given that the solver fails to accurately reflect the network behavior with switching. On the other hand, the network operation with large voltage violations is infeasible as it leads to immediate network collapse. To discourage the agent from pursuing actions that result in actions leading to invalid states, the reward is augmented with a penalty term, $V_{\text{viol}}$. The goal here is to maintain the voltage levels within an acceptable range, ensuring that the network operation is sustainable. The voltage violations for each bus $i \in \mathbf{N}$ beyond its upper limit ($\overline{V}$) and lower limit ($\underline{V}$) are evaluated after power flow estimation as follows:

$$\Delta V_{\max}^i = \begin{cases} \sum_{j \in \phi} V_j^i - \overline{V}, & \text{if } V_j^i > \overline{V} \\ 0, & \text{otherwise} \end{cases} \quad (2)$$

$$\Delta V_{\min}^i = \begin{cases} \sum_{j \in \phi} \underline{V} - V_j^i, & \text{if } V_j^i < \underline{V} \\ 0, & \text{otherwise}. \end{cases} \quad (3)$$

where $\phi$ denotes the set of phase connections for the bus. The voltage measurements and the energy supplied are estimated in per-units (pu) and calculated with respect to the base voltage, $kV_{\text{base}}$, and base power $MVA_{\text{base}}$ of the corresponding network. The per-unit calculations in power systems eliminate the issue of units and is equivalent to normalizing them using their base values:

$$V_{\text{viol}} = \frac{\sum_{i \in \mathbf{N}} (\Delta V_{\max}^i + \Delta V_{\min}^i)}{3|\mathbf{N}|}, \quad (4)$$

where $|\mathbf{N}|$ is the cardinality of the set of network buses, and $\Delta V_{\max}$ and $\Delta V_{\min}$ represent the violations over maximum and minimum desirable voltage limits, respectively.

The outage management tool is applied to power distribution networks where the distribution system operator (DSO) or substation agents are responsible for regulating the power balance and controlling the resources to ensure safe and stable operation. In this study, the test feeders under consideration feature a single substation supplying power to loads while integrating distributed energy resources. Consequently, we adopt a centralized approach for outage management, treating the DSO or substation agent as an autonomous decision-making entity.

The formulation of our approach for outage management is tailored to align with the control architecture found in real-world distribution networks, instead of defaulting to a decentralized approach. Besides this, a multi-agent system (MAS) based approach may prove unsuitable for reconfiguration which relies on wide-area measurements, especially in networks where observability is limited, and local information is constrained. Additionally, the MAS while computationally efficient, encounters challenges in consistently achieving the optimal results[49]. On the other hand, the developed GCAPS with centralized control can achieve near-optimal results by integrating global (wide-area) and local properties into the learning model. It is crucial to highlight that the primary focus of this study does not revolve around designing an MAS architecture, as seen in other works[50,51]. Our objective is not to prescribe the control flow within the smart grid, and we

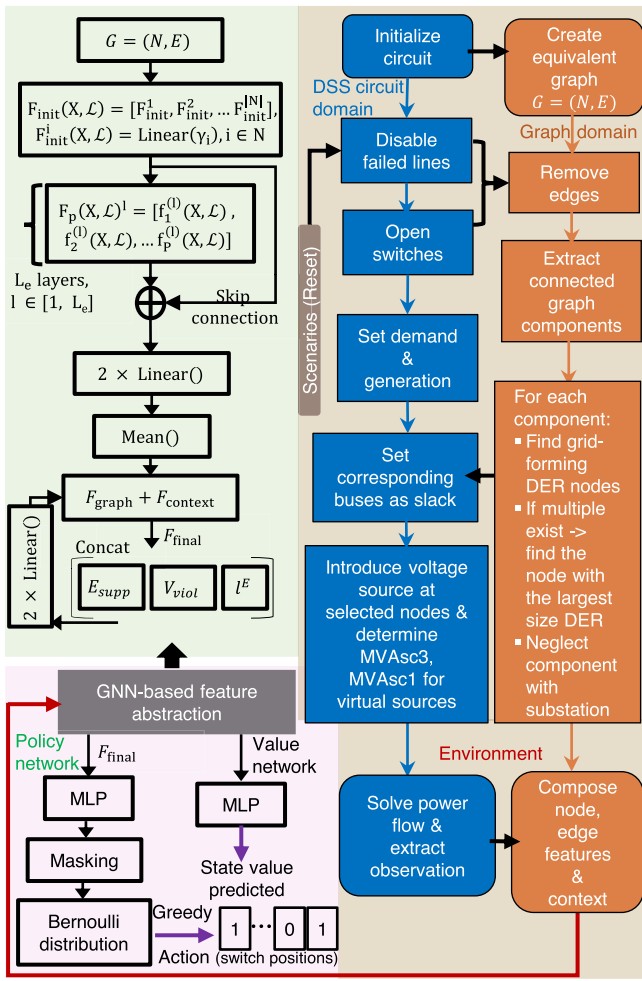

**Fig. 2 | The learning framework developed which includes the environment and the policy network architecture with graph neural network (GNN) based feature abstraction.** The environment is composed of a distribution network modeled using OpenDSS with a Python-based API depicted as the blue-colored blocks. The graph replica of the DSS circuit and the corresponding evaluations in the graph domain are represented using orange-colored blocks. Voltage sources are introduced at virtual slacks specifically for intentional islanding scenarios. The policy network uses the state information from the environment to compute graph node embeddings and context node embeddings using GNN and feedforward networks, respectively. Both the policy and value networks encompass the RL network module, color-coded in light red. The GNN-based graph abstraction is further elaborated in the green section. A final feature vector that encompasses the two embeddings is computed by a multi-layer perceptron (MLP).

operate under the assumption that the existing control architecture, with a DSO (in this case, an autonomous agent), is already established. While acknowledging the evolving nature of control architectures in smart grids, with a potential shift toward distributed control, it is essential to note the current lack of clear standards in this domain.

## Environment and learning architecture

The distribution network models are implemented and simulated using the open-source distribution system simulator (OpenDSS)[52]. DERs are modeled using a generic generator and solar photovoltaic (PV) elements in OpenDSS. Switches are defined on lines with associated switching controls, while the disable/enable property of the loads is used for shedding or picking up load. OpenDSSDirect[53] is employed as the Python-based API to maneuver circuit modifications, I/O operations, and network topology extraction. The equivalent graph is constructed for the circuit using the NetworkX module. The

overall framework of the environment is presented in Fig. 2. The implementation of specific switching actions may lead to the formation of multiple components within the network. These components are then translated into isolated DN sections within the DSS circuit. Furthermore, intentional islands created by grid-forming DERs are considered a potential solution to tackle outages. To enable power flow evaluation in the isolated DSS circuit section, a virtual slack or reference bus is defined at the location of the grid-forming DERs. This requires assigning a voltage source element to the selected buses (i.e., nodes).

The learning architecture utilizes a policy gradient-based GRL algorithm, where the policy network is derived from a Graph Neural Network (GNN). Each node $i$ in the DN graph has properties such as active/reactive power demand, generation, and three-phase voltage measurements, denoted as $\gamma_i = [P_d^i, Q_d^i, P_g^i, Q_g^i, V^i]$. The policy network takes the state information as input and produces an action. The policy network consists of three main components: (1) A GNN which is used to compute the graph node embeddings for the DN graph. (2) A feedforward network that is used to compute a feature vector, referred to as context embedding. This vector incorporates information that cannot be naturally represented in the graph structure, such as the energy supplied, voltage violations, and power flow through the edges. (3) An MLP that takes the node embeddings from the GNN and the context embeddings from the feedforward network as input. It computes a final feature vector that encompasses the entire state space information. Figure 2 shows the overall structure of the policy network, which includes the GNN-based feature abstraction.

Initially, the node properties $\gamma_i, i \in N$, are projected to a higher-dimensional space using linear transformation: $F_{init}^i = W_{init} \times \gamma_i + b_{init}$, where $W_{init} \in \mathbb{R}^{|\gamma_i| \times h_0}$ and $b_{init}$ are learnable weights and biases, respectively. The cardinality of a vector or set is denoted by $|.|$, and $h_0$ represents the projection length. Let $F_{init}$ be a matrix ($\in \mathbb{R}^{|N| \times h_0}$) that represents all $F_{init}^i, i \in N$, ($F_{init} = [F_{init}^1, F_{init}^2 \ldots F_{init}^{|N|}]$)

Node embeddings: Each feature vector $F_{init}^i, i \in \mathbf{N}$, is then passed through a series of Graph capsule layers. These layers utilize a graph convolutional filter of polynomial form to compute a matrix $f_p^{(l)}(\mathcal{X}, \mathcal{L})$, defined as:

$$f_p^{(l)}(\mathcal{X}, \mathcal{L}) = \sum_{k=0}^{K} \mathcal{L}^k \left( F_{(l-1)}(\mathcal{X}, \mathcal{L})^{\mathbf{p}} W_{pk}^{(l)} \right). \tag{5}$$

Here, $\mathcal{L}$ represents the graph Laplacian, $p$ is the order of the statistical moment, $K$ is the degree of the convolutional filter, $F_{(l-1)}(\mathcal{X}, \mathcal{L})$ denotes the output from layer $l-1$, and $F_{(l-1)}(\mathcal{X}, \mathcal{L})^{\mathbf{p}}$ represents $p$ times element-wise multiplication of $F_{(l-1)}(\mathcal{X}, \mathcal{L})$. Here, $F_{(l-1)}(\mathcal{X}, \mathcal{L}) \in \mathbb{R}^{N_n \times h_{l-1}p}$, $W_{pk}^{(l)} \in \mathbb{R}^{h_{l-1}p \times h_l}$. The variable $f_p^{(l)}(\mathcal{X}, \mathcal{L}) \in \mathbb{R}^{N_n \times h_l}$ is a matrix, where each row is an intermediate feature vector for each node $i \in \mathbf{N}$, infusing nodal information from $L_e \times K$ hop neighbors, for a value of $p$. The output of layer $l$ is obtained by concatenating all $f_p^{(l)}(\mathcal{X}, \mathcal{L})$, as given by:

$$F_l(\mathcal{X}, \mathcal{L}) = \left[ f_1^{(l)}(\mathcal{X}, \mathcal{L}) f_2^{(l)}(\mathcal{X}, \mathcal{L}), \ldots f_{\mathcal{P}}^{(l)}(\mathcal{X}, \mathcal{L}) \right]. \tag{6}$$

Here, $\mathcal{P}$ is the highest order of statistical moment, and $h_l$ is the node embedding length of layer $l$. We consider all the values of $h_l, l \in [0, L_e]$, to be the same throughout the paper. Equations (5) and (6) are computed for $L_e$ layers, where each layer uses the output from the previous layer ($F_{l-1}(\mathcal{X}, \mathcal{L})$). Increasing the number of layers ($L_e$) and raising the value of $K$ can enhance the learning of the overall structure of the graph by aggregating nodal neighborhood features from $L_e \times K$ neighbors. However, this improvement comes at the expense of having more learnable parameters in the policy, which becomes a drawback as the problem size increases. A larger value of $h_l$ is beneficial as it enables the computation of a more detailed and comprehensive nodal state representation, both at the final stage and in intermediate steps.

Similarly, a larger value of $P$ assists in a better encoding of intermediate states using a vector representation (described in Eq. (6)) for each intermediate feature. This richer structural embedding is expected to be more effective than the scalar embedding used in GCN (Graph Convolutional Networks). However, it is important to note that both higher $h_l$ and $P$ come with additional training costs. The final node embeddings are computed using a linear transformation of $F_{l=L_e}(\mathcal{X}, \mathcal{L})$:

$$F_{Nodes} = F_{l=L_e}(\mathcal{X}, \mathcal{L}).W_F, \tag{7}$$

where $W_F$ is a learnable weight matrix of size $h_{L_e}\mathcal{P} \times h_{L_e}$.

The final graph embedding is computed by passing the node embeddings matrix $F_{Nodes}$ through a series of Linear layers, followed by taking the mean:

$$F_{graph} = Mean(W_{g2} \times (W_{g1} \times F_{Nodes})), \tag{8}$$

where $W_{g1} \in \mathbb{R}^{h_{L_e} \times |N|}$ and $W_{g2} \in \mathbb{R}^{h_{L_e} \times h_{L_e}}$, and $F_{graph} \in \mathbb{R}^{h_{L_e}}$, for ease of representation, the bias terms are omitted here.

Context: In addition to the graph-based information, certain state space variables cannot be directly represented as nodes in the graph. These variables include energy supplied $E_{supp}$, voltage violation $V_{viol}$, and power flow through the edges $l^E$. The measurement of the impact of a control action on the distribution network performance serves as the context for training the model to embrace control policies that are both operationally feasible and safe. In the case of power networks, voltage violations can lead to severe consequences. The objective during steady-state operation is to uphold network voltage to prevent under-voltage and the ensuing blackout. Additionally, switching induces alterations in the network state, consequently causing a shift in the supplied energy. This impact is also considered as contextual information for the learning model. Similarly, the power flow through the branches which is representative of the DN state and line status (on, off, or outage) is encompassed within the context. To incorporate this information, a feature vector called the context is constructed:

$$F_{context} = Feedforward(Concat([E_{supp}, V_{viol}, l^E])). \tag{9}$$

Final MLP layer: The final state embedding $F_{final}$ ($\mathbb{R}^{h_{L_e}}$) is computed by adding $F_{graph}$ and $F_{context}$ and passing it through an MLP layer:

$$F_{final} = MLP(F_{graph} + F_{context}). \tag{10}$$

The Logits $\in \mathbb{R}^{|\mathcal{A}|}$ across all available actions are computed by passing $F_{final}$ through a Feedforward layer. The Logits of the switches that need to be masked are set to negative infinity. Using the Logits, a Bernoulli probability distribution is computed for all available actions, with the probabilities computed using a Sigmoid function as $e^{Logits}/(1 + e^{Logits})$. The final switching action is determined using a greedy policy. If the mean of an action element (switch) is greater than 0.5, the switch position is set as on (or a value of 1).

The predicted value of the state is computed by passing $F_{final}$ through another feedforward layer, which approximates the value of the state.

For this policy to be implemented on power networks of different sizes, the only change that has to be made is in the Feedforward layer used to compute the "context" vector. This is because the Feedforward layer size depends on the size of the state variables $l_E$ and $E_{supp}$, which varies with the power network size. The structure of the GCAPS encoder and the final MLP layer does not need to change, hence the GCAPS encoder and the final MLP layer trained for a smaller-sized network, could also be used as a warm start to train for a larger-sized network. This is a significant fundamental advantage of the choice of our GNN architecture used to embody the network reconfiguration policy.

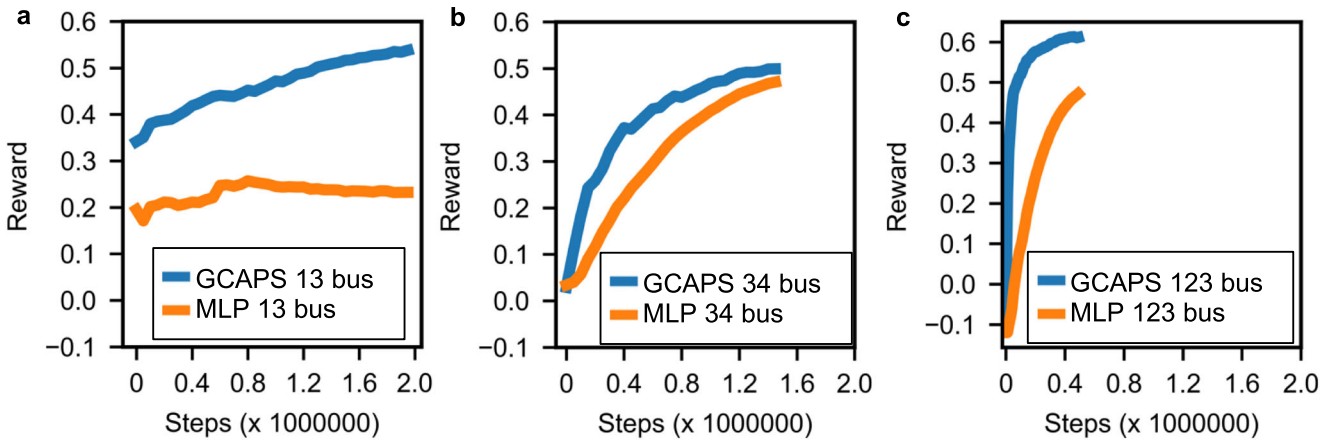

**Fig. 3 | The training convergence plots for the policy models.** Our GCAPS-based GRL model is compared with the MLP model which does not utilize graph abstraction. **a** Convergence plot of GCAPS and MLP for the 13-bus network. **b** Convergence plot for the 34-bus network. **c** Convergence plot for the 123-bus network. Source data are provided as a Source Data file.

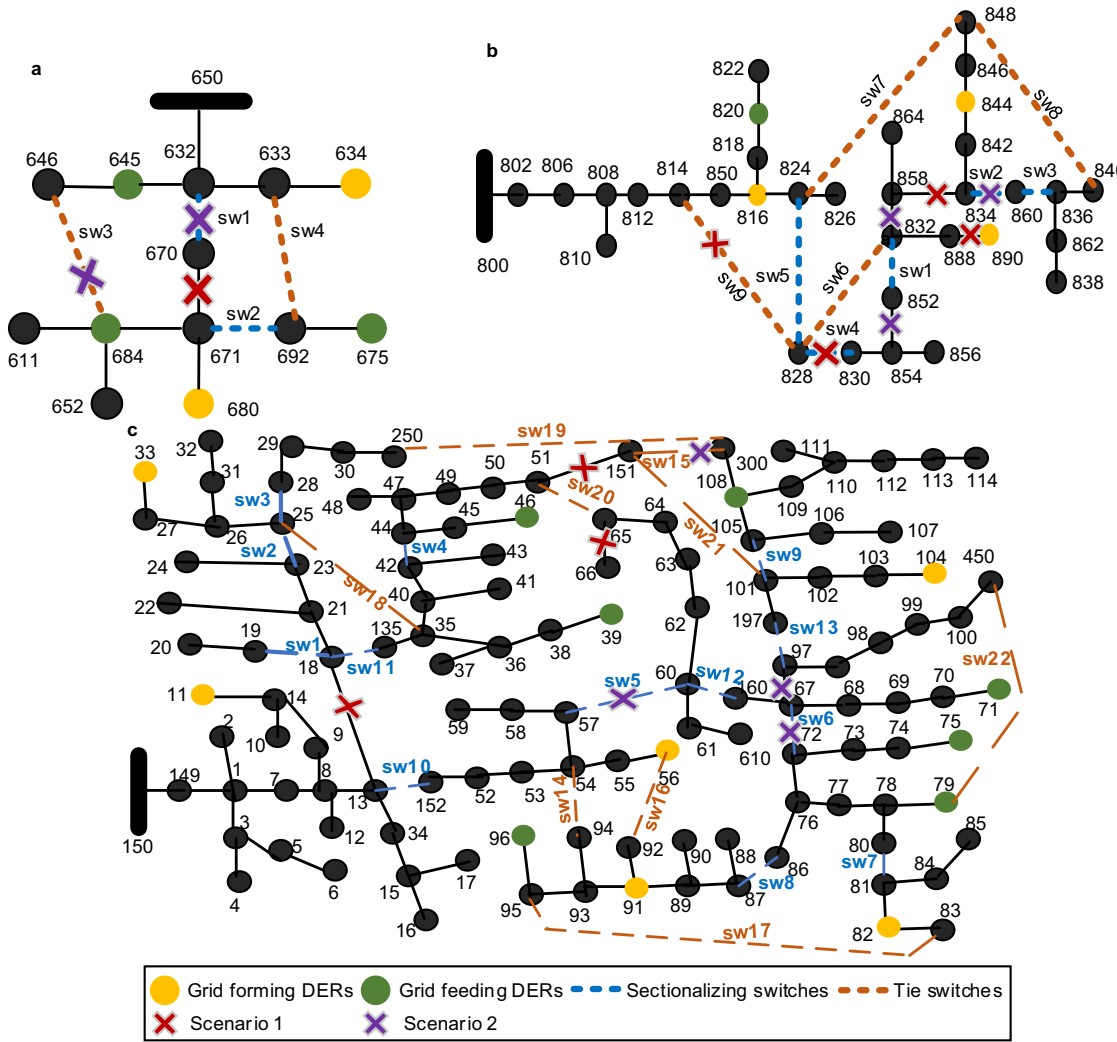

**Fig. 4 | Test networks used to validate the proposed GCAPS model for real-time resilient control.** The test networks are IEEE distribution networks modified by the addition of switches and DERs. Switches in the networks include sectionalizing and tie switches. Both types of DERs, i.e., grid-forming and grid-feeding, are considered. **a** Modified 13-bus IEEE test network. **b** Modified 34-bus IEEE test network. **c** Modified 123-bus IEEE test network. The two specific test scenarios used in each network are also marked.

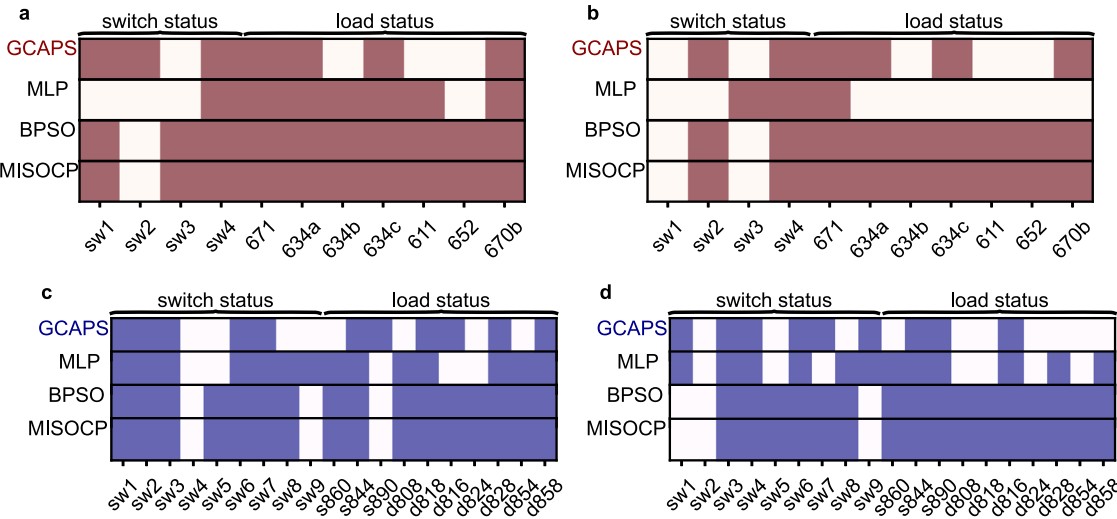

**Fig. 5 | Status of decision variables acquired from the proposed model and baselines for the 13 and 34 bus test networks.** The light entries denote 0 indicating an open status while the darker entries denote 1 representing a closed status. **a** Status of decision variables for test scenario 1 in 13-bus network with line outage 670–671. **b** Status of decision variables for test scenario 2 in 13-bus network with line outages 632–670, and 646–684. **c** Status of decision variables for test scenario 1 in 34-bus network with line outages 858–834, 888–890, 814–828, and 828–830. **d** Status of decision variables for test scenario 2 in 34-bus network with line outages 832–858, 854–852, and 834–860. Source data are provided as a Source Data file.

## Training process

The training process involves generating samples on the distribution network to simulate different outage scenarios. This is accomplished by introducing line failures, adjusting load and generation operating points, and considering various outage scenarios. The outage events in the network are primarily caused by distribution line failures, which are simulated using a graph-based approach (discussed in the "Methods" section). The power network operating points (i.e., the load demand and power generation) are randomly drawn out of an annual profile made available in OpenDSS. To train the policy network, we employ Proximal Policy Optimization (PPO)[54]. Here, the PPO training algorithm has been implemented using the stable-baselines3[55] python library. On-policy algorithms such as PPO are usually preferred over off-policy algorithms for environments with a discrete action space. An additional advantage of using the PPO implementation is its ability to support all the available data types in stable-baselines3[55]. This is particularly useful to address the action space in our problem, which is represented in terms of MultiBinary data type. The training process involves collecting experience in the form of tuples containing the state, action, reward, and the next state. PPO operates based on rollout operations, where each operation consists of a fixed number of steps, denoted as $N_{steps}$. The weight updates occur after completing a rollout operation, in batches of size $N_{batch}(\leq N_{steps})$. The weight update is performed via backpropagation, aiming to minimize a cost function comprising the policy gradient loss and the state value approximation loss. The policy network was trained for a total of $N_{total}$ number of steps. To evaluate the performance of the proposed model, as well as to assess the impact of local and global structural information in the encoding process, we conducted comparative experiments with another learning-based framework called MLP. This framework utilizes the PPO algorithm, with a policy network based on a simple Multi-Layer Perceptron (MLP) architecture. To ensure a fair and unbiased comparison, MLP was trained using the same settings as GCAPS. Both the MLP-based policy and the GCAPS-based policy are trained on an Intel Xeon Gold 6330 CPU (including 28 cores) with 512GB RAM and an NVIDIA A100 GPU. Note that this is expected to be a one/few-run offline investment for any given or existing network. Moreover, such (or even better) computing resources are readily available nowadays, making the training process a reasonable offline investment for

training a real-time decision-support system (the policy models) for outage management. This solution strategy is particularly attractive considering that the real-time models are much faster than current baselines, as seen from the comparisons with baselines in the "Results" section.

Figure 3 shows the training history in terms of the average episodic reward after each rollout, while training GCAPS and MLP for 13, 34, and 123 bus systems. The average episodic reward is computed as the average of the episodic rewards for all the episodes in each rollout operation. Analyzing the training history curve depicted in Fig. 3, it becomes evident that GCAPS consistently achieves a higher reward compared to MLP for the 13-bus, 34-bus, and 123-bus systems. For the 13-bus network, the average episodic reward for MLP converges to a slightly lower value than the peak value, while for GCAPS, the average episodic rewards are much higher compared to MLP, but could not fully converge in 2 million steps. For 34-bus and 123-bus networks, GCAPS has a faster convergence compared to that of MLP. This observation demonstrates the superior performance of GCAPS in effectively managing outages and optimizing the distribution network's operational state. The codes for training can be found in ref. 56.

## Case study on 13-bus network

The proposed model for outage management is validated using a modified version of the IEEE 13-bus distribution test network. This network incorporates switches and DERs and serves as the basis for validating the effectiveness of the proposed model, as shown in Fig. 4a. The quantity, positions, and specifications of the switches within the 13-bus test network are based on established studies that have previously validated the technical viability of these components within the circuit. Specifically, for the 13-bus network, we refer to the details presented in refs. 57,58 to define the sectionalizing and tie switches. Our model assumes that switches are pre-installed in the network with their data available for our decision-making tool. However, optimizing switch locations and quantities falls within a planning study and requires a techno-economic analysis, which is beyond the scope of this paper. Our focus is on evaluating the model for enhancing operational resilience in power networks. Two grid-forming DERs of 1000 kW are considered at buses 634 and 680, while the buses 645, 675, and 684 are equipped with grid-feeding DERs rated at 40 kW, 500 kW, and 100 kW,

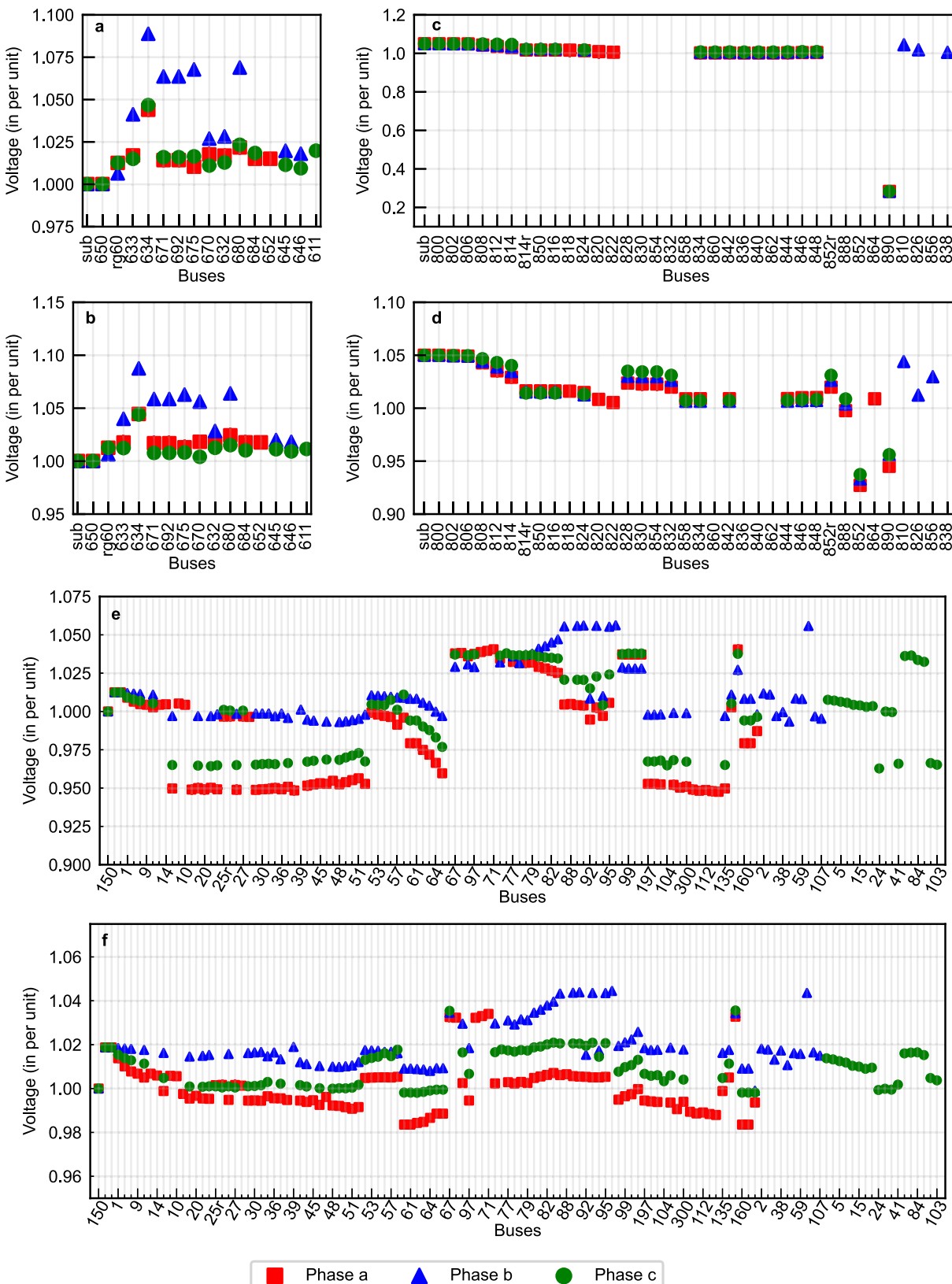

**Fig. 6 | Voltage plot of the test networks with the GCAPS outage management solution implemented during outages for test scenarios.** Only the active phases of the buses in the network are marked in the plot. **a** Test scenario 1 in 13-bus network. **b** Test scenario 2 in 13-bus network. **c** Test scenario 1 in 34-bus network. **d** Test scenario 2 in 34-bus network. **e** Test scenario 1 in 123-bus network. **f** Test scenario 2 in 123-bus network. Source data are provided as a Source Data file.

respectively. The total connected load of the network is 3.5 MW. In the normal configuration of the network, the sectionalizing switches are closed, while the tie switches remain open. This initial setup establishes the baseline operational state for the network. To systematically evaluate the developed model and its performance, two traditional optimization techniques, namely the mixed integer second-order conic programming (MISOCP) and binary particle swarm optimization (BPSO), are employed for all case studies in addition to the previously discussed MLP model. In the testing phase of the models, we rationally select the number and location of the line outages as opposed to the graph-based approach used during training. Additionally, the load and generating points are not drawn out of the representative annual profile discussed in training, rather a randomly generated multiplying factor is used to set the network operating point.

Scenario 1 in the 13-bus network involves the failure of a single line of importance, determined by its high edge-betweenness in normal configuration. Specifically, this scenario represents the outage of the line connecting buses 670–671. The status of the decision variables, which includes both the switches and dispatchable loads, obtained from the different models for scenario 1 is depicted in Fig. 5a. Notably, both the traditional optimization models, namely the MISOCP and the BPSO, yield the same solution for scenario 1. An important observation from analyzing the statuses of the switches and loads is that the reinforcement learning models demonstrate generalizability by providing distinct solutions for the two different scenarios. It is worth mentioning that the MLP model generates different solutions for the same test case while the GCAPS model solution is reproducible for a specific test case. The voltage plot of the 13-bus network, after implementing the GCAPS solution for managing outage scenario 1, is illustrated in Fig. 6a. The GCAPS solution reroutes the power from the substation to affected downstream section through an alternate path. Due to this switching action in scenario 1, the resulting network configuration maintains a robust connection to the substation, ensuring that the voltages at all active phases of connected buses are within 0.99 and 1.10 pu, thus operating well within the desirable bounds.

Scenario 2 involves the outage of two switchable lines connecting 632–670, and 646–684. This scenario aims to test the capability of the proposed model to enforce the inoperability of the outage switch in decision support. The status of decision variables, including the switches and dispatchable loads, obtained from the different models for scenario 2, is shown in Fig. 5b. Once again, the MISOCP and the BPSO solutions for scenario 2 are identical. Upon inspecting the decision variables, it is noticeable that the MLP-based RL model violates the non-switchable condition of the outage line 646–684 (sw3) for scenario 2, as it mistakenly closes the switch. The voltage plot of the 13-bus network, after implementing the GCAPS solution for managing outage scenario 2, is shown in Fig. 6b. In scenario 2, the GCAPS outage mitigation solution ensures a functional network with voltages at buses ranging from 1.10 pu to 0.99 pu. This solution also does not isolate any components of the network from the substation, thereby resulting in a stronger connected network. Additionally, the diversity in solutions with different outage scenarios is indicative of the generalizing capability of the model.

## Case study on 34-bus network
The validation of the proposed model and baselines is conducted on a modified 34-bus distribution test network, which incorporates switches and DERs. The details regarding the switches in the 34-bus network are adopted from ref. 59, albeit presented in a different ordering of sectionalizing and tie switches here. The total connected load of the network is 2.04 MW. Three grid-forming DERs with capacities of 146 kW, 144 kW, and 200 kW are connected at buses 890, 844, and 816, respectively while a grid-feeding DER with a capacity of 96 kW is connected at bus 820 as shown in Fig. 4b. Under normal operating

conditions, the five sectionalizing switches are closed, while the four tie switches are open.

Scenario 1 involves multiple line outages at the connections between buses 858–834, 888–890, 814–828, and 828–830. The lines connecting the buses 814–828 and 828–830 are switchable lines (switches 9 and 4, respectively). While the line 858–834 is one with a high edge betweenness centrality measure in the downstream section of the feeder. Figure 5c presents the status of the decision variables, including switchable lines and loads, obtained from the different models for scenario 1. Both the MISOCP and BPSO yield similar results for scenario 1 on the 34-bus network. The results demonstrate the ability of RL models to differentiate between various scenarios and generalize during decision-making. However, the MLP-based RL model produces an invalid control action in scenario 1 by closing switch 9 on the outage line. The switching action from the GCAPS forms two network components. One is connected to the substation and hence the voltage measurement at these buses are within the desirable limits as seen in Fig. 6c. The other network section is formed around the DER at bus 890. However, this DER is not a grid-forming DER and therefore, the loads at these buses remain unsupplied. This is observed by the inactive or zero voltage for certain buses in the voltage profile plot (Fig. 6c). As shown in the figure, the voltage at bus 890 violates the safe operational limits. However, this is because of the grid-feeding DER at the bus 890. The grid-feeding DERs are generally equipped with island detection modules that turn off the DER when isolated. The voltages at all the other active buses are found to be within the limits of 0.95–1.10 pu.

Scenario 2 considers multiple line failures at 832–858, 834–860, and 854–852 in the network. The lines 832–858 and 854–852 are in close proximity, while the line 834-860 is a switchable sectionalizing line (switch 2). Figure 5d presents the status of the decision variables, including the switchable lines and loads, obtained from the different models for scenario 2. In scenario 2, the switching action by the GCAPS model results in a configuration that remains connected to the substation, with a small section disconnected (inactive) from the main network. The voltage plot for the 34-bus network, derived by implementing the GCAPS solution for scenario 2, is presented in Fig. 6d. The buses disconnected from the network by the switching action are characterized by inactive (or zero voltage from OpenDSS) as seen in Fig. 6d. It is observed that the GCAPS solution for scenario 2 ensures voltages at all active phases of connected buses are well within the range of 0.90–1.10 pu.

## Case study on 123-bus Network
To assess the scalability of the proposed learning over graphs model, we applied the developed outage management tool to a modified IEEE 123-bus test network. This network has been modified by the inclusion of 13 sectionalizing and 9 tie switches as shown in Fig. 4c. The specifications of the switches are obtained from ref. 58, albeit with a different arrangement in our implementation. The DERs with a capacity of 250 kW are connected at buses 39, 46, 71, 75, 79, 96, and 108, while grid-feeding DERs sized at 80 kW are introduced at buses 11, 33, 56, 82, 91, and 104, as detailed in ref. 60. During normal operating conditions, the sectionalizing switches are in the closed position and the tie switches are open. Two outage scenarios have been considered to test the GCAPS model taking into account the network centrality metrics and associated vulnerabilities.

In scenario 1, outages have been considered on lines connecting buses 13–18, 51–151, and 65–66. Notably, the edge 13–18 exhibits the highest current-flow betweenness centrality, while nodes 51 and 151 have high current-flow closeness centrality. Additionally, the edge 65–66 is located at the end of a lateral feeder section. Figure 7a presents the status of the decision variables including switching lines and loads acquired from the different methods for scenario 1. The MISOCP yields the optimal result. The BPSO here, however does not produce

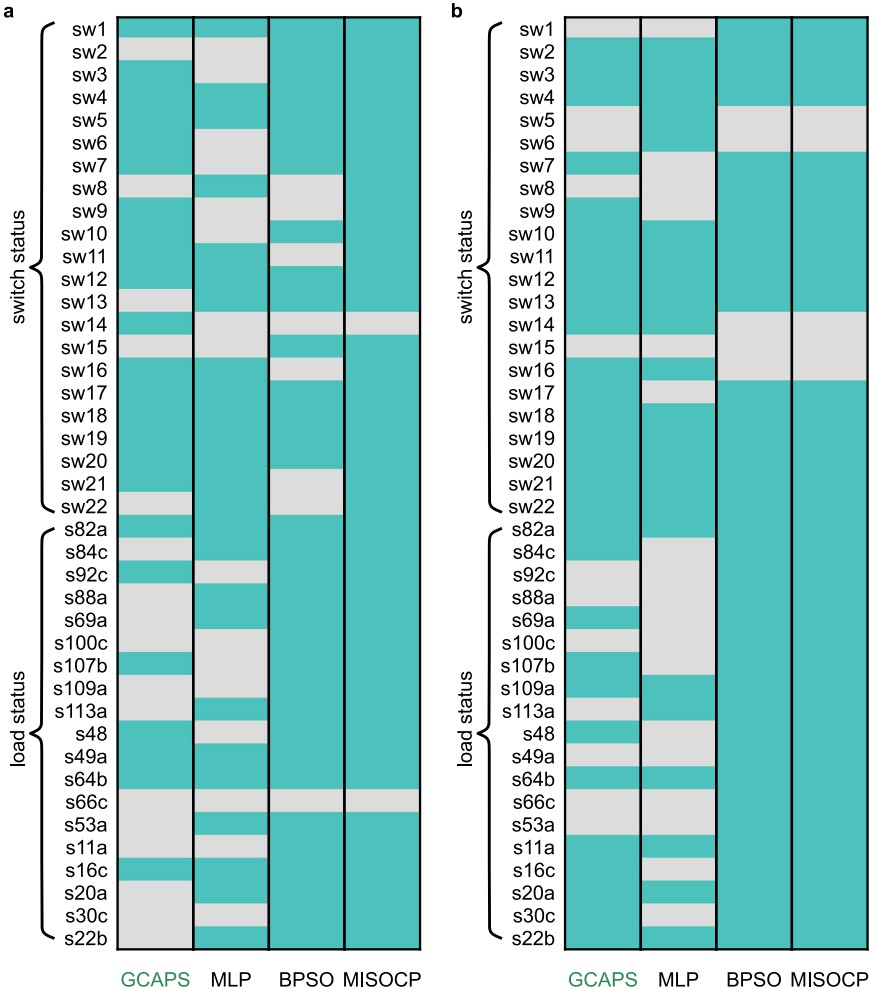

**Fig. 7 | Status of decision variables output from the proposed model and baselines for the 123-bus network.** The light entries denote 0 indicating an open status while the darker entries denote 1 representing a closed status. **a** Test scenario 1 with line outages: 13–18, 15–151, and 65–66. **b** Test scenario 2 with line outages: 151–300, 57–60, 67–72, and 67–97. Source data are provided as a Source Data file.

the same result as MISOCP (as seen in other case studies) and seems to be stuck at a local optimum (clarified in Fig. 8a). There are no invalid switching actions in this scenario. The GCAPS switching action when implemented on the network suffering from an outage, results in improved performance with voltage profile as shown in Fig. 6e. The phases disconnected by switching and inactive phases are indicated as 0 when evaluating the network circuit in OpenDSS. Hence, the voltage measured at the active phases of all the buses are plotted in Fig. 6e. It is observed that the bus voltages are well within the desirable limits following outage management by GCAPS.

In scenario 2, multiple outages at lines connecting buses 151–300, 57–60, 67–72, and 67–97 are considered, and among these, the first three lines are associated with switches (sw15, sw5, and sw6 respectively). The last line connects end nodes with high betweenness centrality. Figure 7b illustrates the status of the decision variables, encompassing switchable lines and loads output by different models for scenario 2. The MLP model is found to operate outage switches, thus producing invalid actions. The results for the two outage scenarios in the 123-bus network exhibits the ability of the proposed GRL model to differentiate between scenarios and generalize during decision making. The GCAPS solution on the 123-bus network with outages results in improved network performance and the corresponding voltage plot is displayed in Fig. 6f. As seen in the figure, for the specific case, the voltage at the buses (for active phases) are within desirable bounds using the GCAPS switching control.

## Comparison of the proposed model with baselines

We compare the developed GCAPS-based GRL model with the baseline models to evaluate the performance and the estimated energy served during outage conditions. Figure 8a, b presents the estimated equivalent energy served when implementing the control decisions in the distribution test networks for scenarios 1 and 2 using the different models, respectively. In the 13-bus network, as expected, the energy supplied is optimal for the MISOCP and BPSO models. Our GCAPS model shows near-optimal decision-making capability for both scenarios. In scenario 1, the MLP model is inferior as it provides the minimum energy supply among all the models, while it becomes invalid in scenario 2 due to the operation of the outage switch. In the case of 34-bus network our GCAPS model exhibits near-optimal performance, closely approaching the optimal energy supply estimated by the MISOCP and BPSO models. On the other hand, the MLP model performs inferiorly compared to the other models and also produces an invalid control action for scenario 1. As observed in the figure, for the 123 bus network the MISOCP generates the optimal results while the BPSO is near optimal in scenario 1 and optimal in scenario 2. The GCAPS solution closely approaches the optimal solution produced by the exact method. Conversely, the MLP model performs inadequately and results in invalid control actions in scenario 2.

The performance of our GCAPS model is compared with the baselines by testing different scenarios in 13, 34, and 123-bus

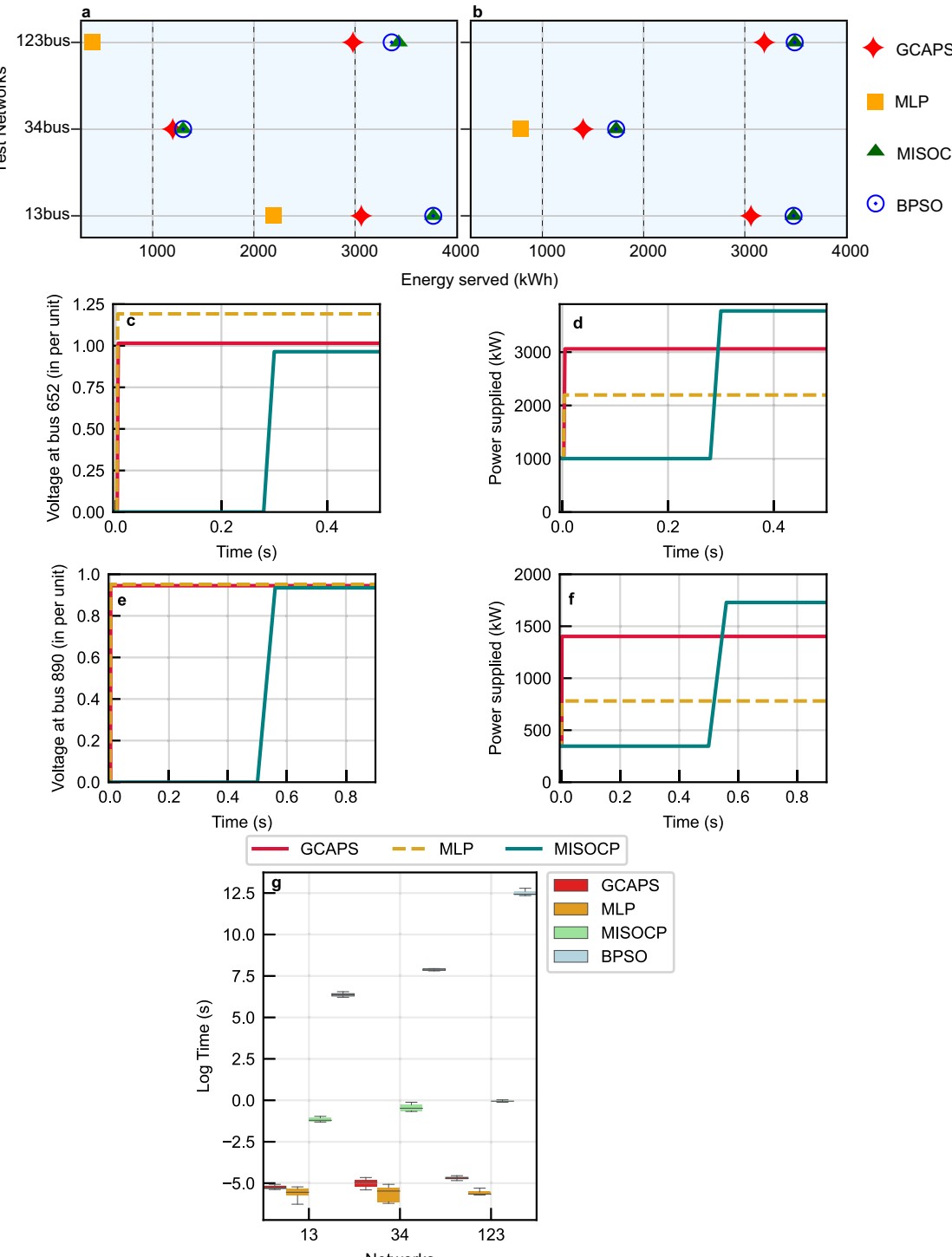

**Fig. 8 | Comparison of the resilience improvement for different models with varying outage scenarios.** The resilience improvement is measured in terms of equivalent energy served when implementing control actions during outages. **a** Energy served by solutions from different models for the three test networks considering scenario 1. **b** Energy served by models for networks in scenario 2. MLP solution is missing for 34-bus in (**a**), and 13 and 123-bus networks in (**b**) due to invalid operation of the outage switch. **c** Voltage measured at bus 652 is used as an indicator for 13-bus network operational characteristics in outage scenario 1. **d** Equivalent power supplied in the network evolving with switching decisions for 13 bus in outage scenario 1. **e** Voltage measured at the bus 890 is used for comparing 34-bus network operational limits in outage scenario 2. **f** Equivalent power supplied in the 34-bus network by different models for scenario 2 and their response with time is illustrated. The baseline models include an MLP-based model which does not utilize graph abstraction and the (slower to compute) conventional mathematical programming technique (here MISOCP). The network is in the disrupted state at time instant 0. **g** Comparison of the performance of the different models across the three test networks. A logarithmic scale of time is adopted to accommodate diverse computational ranges with 5 test runs for each case. Source data are provided as a Source Data file.

**Table 1 | Performance comparison of different models for scenarios in the test networks**

| Network | Method | Scenario 1 Mean time (s) | Scenario 2 Mean time (s) |
|---|---|---|---|
| 13-bus | GCAPS | 0.0049 | 0.0056 |
| | MLP | 0.0039 | 0.0054 |
| | BPSO | 500.15 | 540.20 |
| | MISOCP | 0.3040 | 0.2318 |
| 34-bus | GCAPS | 0.0030 | 0.0025 |
| | MLP | 0.0022 | 0.0020 |
| | BPSO | 2580.15 | 2540.20 |
| | MISOCP | 0.5668 | 0.5676 |
| 123-bus | GCAPS | 0.0090 | 0.0078 |
| | MLP | 0.0030 | 0.0050 |
| | BPSO | 302301.99 | 253251.16 |
| | MISOCP | 0.91 | 0.918 |

networks. The computation time required to obtain the outage mitigation solution is presented in Table 1. The table reports the mean of 5 test runs for the two scenarios using the models across different networks. It can be observed that the response time for the two RL-based models, namely GCAPS and MLP, is in the order of milliseconds, and is mostly agnostic to the increase in the size of the network from 13 to 34 bus system, demonstrating real-time performance. In comparison, the optimization-based methods, BPSO and MISOCP have a delay in computing those decisions. Specifically, BPSO and MISOCP are respectively about 5 and 2 orders of magnitude more expensive than the learned RL-based policies. Although the computational complexity of the proposed model is contingent on the number of switches, the study in ref. 61 found that the optimal number of remote-controlled line switches is 8 to 9 for a 37-node network and 15 to 22 for a 137-node network. Our research aligns with these findings, as we have considered this when defining switches in the networks (nine sectionalizing and tie switches for 34-bus networks and twenty-two switches for 123-bus networks). This approach closely reflects real-world conditions and constraints, as switches are typically not deployed along all lines within the distribution network.

In Fig. 8c–f, we illustrate the performance of the DN and its evolution with time when implementing the decisions provided by the different models during outages. Specifically, the proposed GCAPS-based GRL model is compared with the MLP-based RL model which does not consider the underlying topology and the MISOCP method (conventionally used for solving such problems). The BPSO despite producing similar results as the MISOCP is not suitable for resilience decision support as is evident from the delayed response shown in Table 1. Outage scenario 1 in the 13-bus network and outage scenario 2 in the 34-bus network are used to exemplify the impact of the model response on DN performance. The excluded scenarios in the two networks are not suitable for comparison owing to the invalid switching decisions provided by the MLP model. As observed in Fig. 8c, e, the voltages at the buses 652 and 890 in the 13 and 34 bus networks respectively are under voltage due to disruption. The voltage violation exists for about 10's of cycles in the 13 and 34 bus DNs when MISOCP is used for decision support. While the RL models mitigate the voltage violation through outage management almost instantaneously. The continued operation of the network in the disrupted state also increases the risk of cascaded failures and widespread blackouts. Meanwhile, the loss of energy due to delayed decision-making by the MISOCP with respect to the GCAPS is 607.45 kWs and 596.52 kWs for 13 and 34 buses respectively. In Fig. 8g, the performance of various models on a logarithmic scale of

time across different test networks is illustrated. Test runs of the models for different networks are performed to collect the computation time. A sample size of 5 is employed here as the computation times for BPSO models are prohibitively large.

## Discussion

We have presented a real-time outage management model for distribution networks based on a reinforcement learning over graphs framework. In our outage management model, we have considered the grid-forming and feeding modes of the DER, and hence both grid-connected and islanding reconfiguration schemes have been incorporated into the solution. The load shedding adopted in the mitigation strategy ensures that the network has operational feasibility and is not vulnerable to voltage collapse. The learning model employs an on-policy RL algorithm and adopts the Graph Capsule (GCAPS) neural networks for integrating information about the DN topology into the learning framework. By leveraging GCAPS neural networks, the model has been shown to effectively integrate nodal properties, and local and global structural information into the learning process.

We have evaluated our model on modified versions of the IEEE 13-bus, 34-bus, and 123-bus distribution test networks, which include distributed energy resources (DERs) and sectionalizing/tie switches. Two traditional models based on MISOCP and BPSO, and the RL with MLP as policy network have been used as baselines to compare the real-time decision-making and network resilience improvement capability, where the energy served under disruption (see Fig. 8) can be perceived as a measure of resilience. The results have demonstrated that the proposed model achieves near-optimal performance in real-time outage management for different networks and outage scenarios. Additionally, the model has been found to effectively capture the DN topology in decision-making as indicated by the improved performance and constraint adherence when compared with the MLP-based approach. Above all, our model has also provided time-sensitive decision support for outage mitigation, thereby making it a suitable self-healing tool in the current smart-grid landscape.

As demonstrated in this paper, the rapid decision-making capability in contrast to traditional methods, is a key strength of our model. Unlike conventional approaches, our model demonstrates real-time response times to increasing network size, making it well-suited for online deployment on large distribution networks. However, it is important to note that dealing with larger networks presents challenges during the training phase, demanding advanced computational resources to adequately train the learning over the graphs model. This limitation is encountered during the offline phase and can be resolved by allocating adequate resources for training considering the benefit of operational resilience. From the results in our prior studies on applying related graph-based GRL for Multi-Robot Task Allocation[46,47], we have found that the computational memory requirement for training on larger graphs (more than 200 nodes) is very high and often hinders the training task. Our prior results[46,47] have demonstrated the capability of the GNN-based policy network to learn policies that can be applied to a larger-sized mostly homogeneous networks with simple near-linear state transitions (without training), while still demonstrating comparable performance with respect to more traditional approaches. More work is required to explore if these advantages will also translate to applications such as the DN topology reconfiguration that involves heterogeneous networks and non-linear flow properties that affect the state transition. It is also crucial to model and evaluate the impact of communication breakdowns on resolving power network outages, since those can be an associated artifact attributed to the natural or anthropogenic hazard that caused the power grid breakdown. This, however, necessitates intricate coupled cyber-physical modeling of the communication network, and

formulation of communication recovery as in ref. [62]. Addressing the modeling and control of coupled communication and power networks as a unified effort poses significant challenges. A potential extension of our work involves modeling the interconnected power and communication networks as multi-layered graphs and evaluating the impact of communication failure on power network recovery.

## Methods

### Graph-based scenario generation

The training scenarios used for GRL model were generated from the graph equivalent of the DN. The failure of the components, such as lines, can be approximated by disconnecting them from the DN[63]. The model developed is not specific to any particular type of extreme weather event, and hence a generalized and intuitive approach is adopted for simulating outages during training. The outages in the DN often originate from localized failures that can lead to cascading effects. To emulate this behavior, a sub-graph method for randomized edge removal is employed, similar to the approach described in ref. [64]. This method involves randomly selecting nodes $N_s \in N$ from the graph representation of the DN, and creating subgraphs centered around these nodes with varying radii $R_s \leq R_{\max}$ (maximum radius). We consider $R_{\max} = \frac{G_{\mathrm{dia}}}{2}$, where $G_{\mathrm{dia}}$ is the diameter of the graph. Within each selected subgraph, a fraction of the edges $F_s \in E$ is randomly removed to simulate the localized impact of contingencies. The fraction of edge failures is gradually increased from 0 to 50%. By varying $N_s$, $R_s$, and $F_s$, scenarios with multi-line failures can be generated for training the model. Furthermore, within each scenario, load multipliers and generating points are varied by randomly selecting multipliers from an annual profile available in OpenDSS package with an hourly resolution.

### Mixed-integer programming formulation

Outage management in an unbalanced distribution network is an optimization problem that combines combinatorial and non-linear nature. The problem can be effectively formulated as an optimal power flow problem, leveraging branch flow equations with angle and conic relaxations as in ref. [19]. The decision variables include switching and load shedding, while the control variables corresponding to power flow are also considered in the problem formulation.

For the distribution network with $\mathbb{L}$ set of loads and $\widetilde{\mathbb{L}}$ set of switchable loads, the active/reactive power consumption with load pickup or shedding is modeled using $\delta^L$ as follows:

$$P_i^L = \begin{cases} \delta_i^L P_i^D & \text{if } i \in \widetilde{\mathbb{L}} \\ P_i^D & \text{otherwise} \end{cases}; \forall i \in \mathbb{L} \tag{11a}$$

$$Q_i^L = \begin{cases} \delta_i^L Q_i^D & \text{if } i \in \widetilde{\mathbb{L}} \\ Q_i^D & \text{otherwise} \end{cases}; \forall i \in \mathbb{L} \tag{11b}$$

where $P_i^D$ and $Q_i^D$ represent the active and reactive power demand of the load $i$, respectively.

On the other hand, considering the set of grid-feeding generators $\mathbb{G}_{fd}$ in the DN, the active and reactive power generation is estimated using:

$$P_{(j,k)}^G = P_{\mathrm{avail}}^G / |G_{ph}|; \forall j \in \mathbb{G}_{fd}, k \in \theta^* \tag{12a}$$

$$Q_{(j,k)}^G = Q_{\mathrm{avail}}^G / |G_{ph}|; \forall j \in \mathbb{G}_{fd}, k \in \theta^* \tag{12b}$$

where $P_{\mathrm{avail}}^G$, $Q_{\mathrm{avail}}^G$ is the total generation power available for the generator with $|G_{ph}|$ number of phase connections, and $\theta^*$ is the set of active phases of the generator, considering $\theta = (a, b, c)$.

The total active and reactive power consumption by loads is constrained by the total generation in the DN as follows:

$$\sum_{i \in \mathbb{L}} P_i^L \leq P_{\mathrm{tot}}^G, \sum_{i \in \mathbb{L}} Q_i^L \leq Q_{\mathrm{tot}}^G \tag{13}$$

The total power generation in the DN is given as follows:

$$P_{\mathrm{tot}}^G = \sum_{k \in \theta} \left( \sum_{j \in \mathbb{G}_{fd}} P_{(j,k)}^G + \sum_{h \in \mathbb{G}_s} P_{(h,k)}^G \right) \tag{14a}$$

$$Q_{\mathrm{tot}}^G = \sum_{k \in \theta} \left( \sum_{j \in \mathbb{G}_{fd}} Q_{(j,k)}^G + \sum_{h \in \mathbb{G}_s} Q_{(h,k)}^G \right) \tag{14b}$$

where in addition to the grid-feeding generators, the set of grid-forming generators $\mathbb{G}_s$, including the substation, are considered.

The power supplied by the grid-forming generators and the substation is constrained to be within its maximum capacity as follows:

$$\sum_{k \in \theta^*} P_{(h,k)}^G \leq \overline{P_h^G}; \forall h \in \mathbb{G}_s. \tag{15}$$

Adopting three-phase branch flow formulations with relaxations as in ref. [19], $\mathcal{V}$ and $\mathcal{I}$ are used to denote the square of voltage and current, respectively. The voltages at all buses except the slack buses are constrained within upper and lower limits as follows:

$$\underline{\mathcal{V}} \leq \mathcal{V}_{r,k} \leq \overline{\mathcal{V}}; \forall r \in \mathbb{B} \setminus \mathbb{B}_s, k \in \theta \tag{16}$$

Here, $\mathbb{B}$ and $\mathbb{B}_s$ denote the set of buses and the set of slack buses in the network, respectively. The voltage square at the substation (or slack) bus on the other hand is equated to 1.04 per unit.

For the set of power delivery elements $\mathbb{E}$, the set of switchable elements (lines) $\mathbb{E}_{sw}$, and the line switch status $\delta^{sw}$, the power flow $P^E$ through the elements are constrained as follows:

$$\underline{P}_{(b,k)}^E \leq P_{(b,k)}^E \leq \overline{P}_{(b,k)}^E; \forall b \in \mathbb{E} \setminus \mathbb{E}_{sw}, k \in \theta \tag{17a}$$

$$\underline{P}_{(l,k)}^E \delta_l^{sw} \leq P_{(l,k)}^E \leq \overline{P}_{(l,k)}^E \delta_l^{sw}; \forall l \in \mathbb{E}_{sw}, k \in \theta \tag{17b}$$

In a similar manner, the reactive power flow $Q^E$ and the square of branch current square $\mathcal{I}^E$ through the elements are also constrained within its limits. The power flow through the outage lines defined in set $\mathbb{O}$ is, however, equated to zero as shown below:

$$P_{(b,k)}^E = 0; \forall b \in \mathbb{O}, k \in \theta \tag{18}$$

The reactive power flow and the square of branch current through outage lines are also equated to zero. The balance of active and reactive power flow through the elements is formulated as follows:

$$P_{(b,k)}^E = \sum_{q \in \mathbb{R}_L(b)} P_{(q,k)}^L - \sum_{w \in \mathbb{R}_G(b)} P_{(w,k)}^G + \sum_{h \in \mathbb{C}(b)} P_{(h,k)}^E \\ + R_{(b,k)} \mathcal{I}_{(b,k)}; \forall b \in \mathbb{E}, k \in \theta \tag{19}$$

$$Q_{(b,k)}^E = \sum_{q \in \mathbb{R}_L(b)} Q_{(q,k)}^L - \sum_{w \in \mathbb{R}_G(b)} Q_{(w,k)}^G + \sum_{h \in \mathbb{C}(b)} Q_{(h,k)}^E \\ + X_{(b,k)} \mathcal{I}_{(b,k)}; \forall b \in \mathbb{E}, k \in \theta \tag{20}$$

where $\mathbb{R}_L(b)$ is the set of loads and $\mathbb{R}_G(b)$ is the set of generators connected to the receiving bus of element $b$. In Eqs. (19) and (20), $\mathbb{C}(b)$ represent the elements that are children elements to $b$.

**Table 2 | Training details**

| Parameter | Value |
|---|---|
| Algorithm | PPO |
| Total steps ($N_{total}$) | 2.0e6 (13 bus), 1.5e6 (34 bus), 5e5 (123 bus) |
| Rollout buffer size ($N_{steps}$) | 5e4 |
| Batch size ($N_{batch}$) | 2e4 (13, and 34 bus), 2e3 (123 bus) |
| Optimizer | Adam |
| Initial learning rate ($lr_{init}$) | 1e−5 |
| Decay rate ($D_R$) | 3 |
| Value coefficient | 0.5 |
| Epochs per training update | 100 |

Additionally, Kirchhoff's voltage equation is modeled as:

$$\mathcal{V}_{(\mathbb{R}(b),k)} = \mathcal{V}_{(\mathbb{S}(b),k)} - 2(\hat{R}_{(b,k)} P^E_{(b,k)} + \hat{X}_{(b,k)} Q^E_{(b,k)}) \\ + \hat{Z}_{(b,k)} \mathcal{I}_{(b,k)}); \forall b \in \mathbb{E} \setminus \mathbb{E}_{sw}, k \in \theta \tag{21}$$

Here, the parameters $\hat{R}$ and $\hat{X}$ denote the element's modified resistance and reactance, respectively. In Eq. (21), $\mathbb{S}(b)$ and $\mathbb{R}(b)$ denote the sending and receiving bus of the element $b$, respectively. For elements (lines) with switch, Eq. (21) is modified to an inequality constraint using the big M method[19] and bound within $-(1 - \delta^{sw}_l)M$ and $(1 - \delta^{sw}_l)M$.

The second-order conic inequality constraint using convex relaxation is formulated as follows:

$$\mathcal{I}_{(b,k)} * \mathcal{V}_{(\mathbb{S}(b),k)} \geq [(P^E_{(b,k)})^2 + (Q^E_{(b,k)})^2]; \forall b \in \mathbb{E}, k \in \theta \tag{22}$$

The objective function maximizes the total power supply in the network with control actions during outages and is formulated as follows:

$$\max. \sum_{i \in \mathbb{L}} P^L_i \tag{23}$$

## Training details

The training process is allocated a maximum of 36 h, and the total number of steps is set to 2 million. For the 13-bus network, both GCAPS and MLP successfully completed the training with 2 million steps. However, for the 34-bus network, MLP could only be trained for 1.5 million steps within a 36-h time frame, while the network for 123-bus systems could only be trained for 500,000 steps. To ensure a fair comparison, we utilize the trained weights of GCAPS and MLP at 1.5 million steps for the 34-bus network, and 500,000 steps for the 123-bus network. Here we implement a squared exponential decreasing learning rate strategy $\rho_t = \rho_{init} \times e^{-(1-t)^2 \times D_R}$, where $t$ represents the fraction of current step to the total number of steps for learning, $\rho_t$ is the learning rate at $t$, $\rho_{init}$ is the initial learning rate, and $D_R$ is the decay rate. We used $\rho_{init} = 1e-5$, and $D_R = 3$. This strategy leads to smoother convergence and likely mitigates getting stuck in local minima. Table 2 shows the training details, including the hyperparameter setting for PPO.

## Simulation setup

The proposed model and all the other baselines are tested on a system with Intel Core i7-1365U 1.80 GHz with 16 GB memory. The OpenDSSDirect API along with Python version 3.9.12, and Networkx version 2.6.3 are used in our simulations. The mixed-integer programming is performed with Gurobipy using a Gurobi optimizer version 9.5.2.

## Reporting summary

Further information on research design is available in the Nature Portfolio Reporting Summary linked to this article.

## Data availability

The figure/table data generated in this study are provided in the Source Data file. Source data are provided with this paper.

## Code availability

Code for this article is available publicly from: https://zenodo.org/records/11188543.

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

## Acknowledgements

This material is based upon work sponsored by the Department of the Navy, Office of Naval Research under ONR award number N00014-21-1-2530 (J.Z., S.C., and Y.G.). Part of this material is also based upon work supported by (while Y.G. serving at) the NSF. The United States Government has a royalty-free license throughout the world in all copyrightable material contained herein. Any opinions, findings, and conclusions or recommendations expressed in this material are those of the author(s) and do not necessarily reflect the views of the Office of Naval Research and the National Science Foundation.

## Author contributions

R.A.J. and S.P. conceptualized the code, conducted experiments, and performed analysis. S.C. supervised the development of the learning framework and J.Z. supervised the power network control and evaluation. Y.G. contributed to discussions and provided supervision of the work. R.A.J. and S.P. drafted the manuscript. S.C., Y.G., and J.Z. edited the manuscript. All authors contributed to manuscript revisions and provided feedback.

## Competing interests

The authors declare no competing interests.
