## [Peer Review File · Nature Communications]

Real-time outage management in active distribution networks using reinforcement learning over graphsREVIEWER COMMENTS

Reviewer #1 (Remarks to the Author):

The paper explores improving the resilience of electrical distribution networks via machine learning, specifically using PPO for training a model called GCAPS, and validates its superior performance in managing outages through case studies on 13-bus and 34-bus networks. However, this reviewer has some major concerns in regard to this manuscript.

- 1) The test system employed in the study appears to be relatively small, limiting the scope for evaluating the model's scalability and robustness. I'm curious how the model would perform if tested on a network with a greater number of buses (>100 for example 123 or 136 buses).
- 2) The paper employs Proximal Policy Optimization (PPO) for training the reinforcement learning model. Could you elaborate on the choice of PPO over other RL algorithms? What advantages does it offer in the context of network reconfiguration? The choice of PPO as the RL algorithm is intriguing, especially given its known instability in some applications. I wonder why the authors did not consider using Soft Actor-Critic, which is generally regarded as a more stable alternative.
- 3) The paper compares the proposed model (GCAPS) with a Multi-Layer Perceptron (MLP) and traditional optimization techniques like MISOCP and BPSO. How do these methods differ in computational complexity and scalability?
- 4) The manuscript mentions that the training process was allocated a maximum of 36 hours. Could you provide more insights into the computational resources used for training? How might the training time affect the practicality of the proposed solution?
- 5) The paper introduces a feature vector called the context, which includes variables such as energy supplied, voltage violation, and power flow through the edges. How critical are these contextual variables in the overall performance of the model?
- 6) In Figure 3(a), it's noteworthy that the performance of GCAPS declines at step 1.25, while MLP shows a rise in reward collection. Similarly, Figure 3(b) reveals that the network's performance deteriorates after 800,000 steps. Given these observations, I wonder why the authors chose not to halt the training before reaching this point. Additionally, similar to the case of the 13-bus network, MLP's performance eventually converges with that of GCAPS. This raises the question: if the number of training steps for MLP was increased, would it eventually outperform GCAPS?
- 7) The paper specifies a rather small learning step size, which raises concerns about the model potentially getting stuck in local minima during training. It might be beneficial for the authors to initially set a higher learning rate for the first 50 epochs and then reduce it by a factor of ten for subsequent phases, specifically the next two sets of 100 and 150 epochs. This suggestion contrasts with the paper's current approach, which maintains a constant learning rate across all 100 epochs.
- 8) Given that the paper's primary contribution lies in the proposed algorithm, one would expect the authors to make the code available for replication and further scrutiny. Unfortunately, this opportunity has not been provided, neither for reviewers nor for the broader readership.

Reviewer #2 (Remarks to the Author):

This paper uses reinforcement learning for outage management in smart grids to improve resilience and lost load. While the topic is interesting, some minor and major concerns need to be addressed. My comments are as follows:

- 1) How do you determine or set the number and location of the switches in the smart grid? It has a significant impact on the performance of the approach and the computational burden. Do you optimize them? If Yes, How?
- 2) Do you consider a penalty to show the forbidden states for the agent?
- 3) Is this method centralized or decentralized? considering one single agent to control all switches together is not preferable. Because, in this case, you will need a strong communication system its

own resilience will be a critical issue, which it seems it has been ignored in this paper.

4) The test (13-bus) system is too small. Using this test case, it is not possible to show the time-efficiency of the methodology.

5) Does this method have the possibility to be applied to large-scale systems?

6) working with one smart grid to control it needs communications with the other neighboring smart grids. Any ideas for it? Please elaborate.

Authors Response to Reviews of

Real-Time Outage Management in Active Distribution Networks Using Reinforcement Learning over Graphs

Roshni Anna Jacob, Steve Paul, Souma Chowdhury, Yulia R.Gel, and Jie Zhang

Authors' Response to Review 1 Comments

Review Comment #1:

The test system employed in the study appears to be relatively small, limiting the scope for evaluating the model's scalability and robustness. I'm curious how the model would perform if tested on a network with a greater number of buses (>100 for example 123 or 136 buses).

Authors' Response:

We thank the reviewer for this comment. Acknowledging the reviewer's suggestion, we have also now included the 123-bus test network in addition to the 13 and 34-bus networks, to validate the model performance. We have also included a section describing the case study on the 123-bus network in the revised manuscript. The following modifications have been made to the manuscript:

2.11 Case Study on 123-bus Network

“To assess the scalability of the proposed learning over graphs model, we applied the developed outage management tool to a modified IEEE 123-bus test network. This network has been modified by the inclusion of 13 sectionalizing and 9 tie switches. The specifications of the switches are obtained from [1], albeit with a different arrangement in our implementation. The DERs with a capacity of 250 kW are connected at buses 39, 46, 71, 75, 79, 96, and 108, while grid-feeding DERs sized at 80 kW are introduced at buses 11, 33, 56, 82, 91, and 104, as detailed in [2]. During normal operating conditions, the sectionalizing switches are in the closed position and the tie switches are open. Two outage scenarios have been considered to test the GCAPS model taking into account the network centrality metrics and associated vulnerabilities.

In scenario 1, outages have been considered on lines connecting buses ‘13’-‘18’, ‘51’-‘151’, and ‘65’-‘66’. Notably, the edge ‘13’-‘18’ exhibits the highest current-flow betweenness centrality, while nodes ‘51’ and ‘151’ have high current-flow closeness centrality. Additionally, the edge ‘65’-‘66’ is located at the end of a lateral feeder section. Figure 1(a) presents the status of the decision variables including switching lines and loads acquired from the different methods for scenario 1. The MISOCP yields the optimal result. The BPSO here, however, does not produce the same result as MISOCP (as seen in other case studies) and seems to be stuck at a local optimum (clarified in Fig. 3). There are no invalid switching actions in this scenario. The GCAPS switching action when implemented on the network suffering from an outage, results in improved performance with voltage profile as shown in Fig. 2. The phases disconnected by switching and inactive phases are indicated as 0 when evaluating the network circuit in OpenDSS. Hence, the voltage measured at the active phases of all the buses are plotted in Fig. 2(a). It is observed that the bus voltages are well within the desirable limits following outage management

by GCAPS.

In scenario 2, multiple outages at lines connecting buses ‘151-300’, ‘57-60’, ‘67-72’, and ‘67-97’ are considered, and among these, the first three lines are associated with switches (sw15, sw5, and sw6 respectively). The last line connects end nodes with high betweenness centrality. Figure 1(b) illustrates the status of the decision variables, encompassing switchable lines and loads output by different models for scenario 2. The MLP model is found to operate outage switches, thus producing invalid actions. The results for the two outage scenarios in the 123-bus network exhibits the ability of the proposed GRL model to differentiate between scenarios and generalize during decision making. The GCAPS solution on the 123-bus network with outages results in improved network performance and the corresponding voltage plot is displayed in Fig. 2(b). As seen in the figure, for the specific case, the voltage at the buses (for active phases) are within desirable bounds using the GCAPS switching control.”

2.12 Comparison of the Proposed Model with Baselines

“Similarly, the estimated equivalent energy served for the two scenarios in the 123-bus network with different models is depicted in Fig. 3. As observed, the MISOCP generates the optimal results while the BPSO is near optimal in scenario 1 and optimal in scenario 2. The GCAPS solution closely approaches the optimal solution produced by the exact method. Conversely, the MLP model performs inadequately and results in invalid control actions in scenario 2.”

In addition, note that, we have also included a small excerpt discussing how the graph neural network (GNN) embodying the network reconfiguration policy must be updated when the network size changes. Since the encoder and decoder, which are key portions of the architectures, does not need to change as the network size increases, it provides a fundamental scalability advantage over typical non-graph based architectures. The added portion is also quoted below.

2.5 Learning Architecture

For this policy to be implemented on power networks of different sizes, the only change that has to be made is in the Feedforward layer used to compute the “context” vector. This is because the Feedforward layer size depends on the size of the state variables l_E and E_{supp} , which varies with the power network size. The structure of the GCAPS encoder and the final MLP layer does not need to change, hence the GCAPS encoder and the final MLP layer trained for a smaller-sized network, could also be used as a warm start to train for a larger-sized network. This is a significant fundamental advantage of the choice of our GNN architecture used to embody the network reconfiguration policy.

Review Comment #2:

The paper employs Proximal Policy Optimization (PPO) for training the reinforcement learning model. Could you elaborate on the choice of PPO over other RL algorithms? What advantages does it offer in the context of network reconfiguration? The choice of PPO as the RL algorithm is intriguing, especially given its known instability in some applications. I wonder why the authors did not consider using Soft Actor-Critic, which is generally regarded as a more stable alternative.

Figure 1: **Status of decision variables output from the proposed model and baselines for the 123-bus network.** The light entries denote '0' indicating an open status while the darker entries denote '1' representing a closed status. (a) Test scenario 1 with line outages: '13-18', '15-151', and '65-66'. (b) Test scenario 2 with line outages: '151-300', '57-60', '67-72', and '67-97'.

Figure 2: **Voltage plot of the 123-bus test network for varying outage scenarios with the GCAPS outage management solution.** Only the active phases of the buses in the network are included in the plot. (a) Outage scenario 1. (b) Outage scenario 2.

Figure 3: **Comparison of the resilience improvement for different models with varying outage scenarios.** (c) Testing on the 123-bus network, with empty bar for MLP model in scenario 2 due to operation of outage switch.

Authors' Response:

We thank the reviewer for raising this question. The choice of the RL algorithm for training our policy model is indeed important. In this regard, it is important to note that among the popular methods, each has its pros and cons, and there isn't strong evidence of one being universally superior to others. For example, although Soft Actor-Critic (SAC) could be in some cases more stable, PPO is often faster in training compared to SAC for multi-agent problems [3], [4]. Moreover, SAC is better suited for continuous action spaces, while for discrete action spaces (as is the case in our problem here), PPO is often the better choice. In addition, there are also a few key implementation/programmatic advantages in using PPO for our use case and policy model – the stable-baselines3 implementation of SAC does not support certain essential data types, while the PPO implementation does support all the data types available in stable-baselines3. For our specific use case, the action space is modeled as a MultiBinary data type, which is not supported by the SAC algorithm in stable-baselines3. In light of this comment, to further clarify the choice of the RL training algorithm, the following additions have been made as quoted below:

2.6 Training Process

“Here, the PPO training algorithm has been implemented using the stable-baselines3 [5] python library. On-policy algorithms such as PPO are usually preferred over off-policy algorithms for environments with a discrete action space. An additional advantage of using the PPO implementation is its ability to support all the available data types in stable-baselines3. This is particularly useful to address the action space in our problem, which is represented in terms of MultiBinary data type.”

Review Comment #3:

The paper compares the proposed model (GCAPS) with a Multi-Layer Perceptron (MLP) and traditional optimization techniques like MISOCP and BPSO. How do these methods differ in computational complexity and scalability?

Authors' Response:

We thank the reviewer for raising this point. The GCAPS model, developed in this paper, is a reinforcement learning (RL) model over graphs used to facilitate switching control decisions during outages. The outage management problem is an NP-hard combinatorial optimization problem [6], which can be solved using classical mixed-integer non-linear programming (MINLP) and metaheuristics optimization techniques. Mixed-integer second-order conic programming (MISOCP) and binary particle swarm optimization (BPSO) are respectively two such optimization-based baselines. It is in practice challenging to derive theoretical measures of their computational complexity, say as a function of the size of network being actively reconfigured. Hence, we use empirical studies to compare their computing time performance and report that in Table 1 in the manuscript. For further reference, we have copied below the table and boxplots (Fig. 4) comparing computing time performance of each method. This computing time reflects the decision-making time required by the different methods when provided with a specific system state inclusive of outage lines, to which they must respond. MISOCP, while achieving optimal results, exhibits a computational expense of approximately two orders of magnitude higher than the RL-based GCAPS model, making it risky for real-time resilience decision making. On the other hand, BPSO is too expensive (mins to hours even on moderate sized networks) to be even practically viable for any sort of online decision-making. These discussion and results illustrations are available in Section 2.12 of the revised manuscript. In addition, as a future work, we also intend to test the models on larger practical networks (Eg., San Francisco Bay Area Synthetic Networks), where the efficiency of the proposed approach may be more evident.

Table 1: Performance comparison of different models for scenarios in the test networks

Network	Method	Scenario 1	Scenario 2
		Mean time (s)	Mean time (s)
13-bus	GCAPS	0.0049	0.0056
	MLP	0.0039	0.0054
	BPSO	500.15	540.20
	MISOCP	0.3040	0.2318
34-bus	GCAPS	0.0030	0.0025
	MLP	0.0022	0.0020
	BPSO	2580.15	2540.20
	MISOCP	0.5668	0.5676
123-bus	GCAPS	0.0090	0.0078
	MLP	0.0030	0.0050
	BPSO	302301.99	253251.16
	MISOCP	0.91	0.918

Review Comment #4:

The manuscript mentions that the training process was allocated a maximum of 36 hours. Could you provide more insights into the computational resources used for training? How might the training time affect the practicality of the proposed solution?

Authors' Response:

We express gratitude to the reviewer for the question. The training process was performed on an Intel Xeon Gold 6330 CPU (28 cores) with 512GB RAM and an NVIDIA A100 GPU, which has now been reported in the revised manuscript in Section 2.6 as quoted below. Such resources are relatively mainstream, and since it is a one-time offline training cost for any given/existing network, it is a reasonable investment to make to develop an online or real-time decision-support system (that is much faster than existing baselines, as seen from our results) for network reconfiguration. In addition, note that, the 36 hours hard stop for the training process was a practical choice, and this choice will depend on the size of the network being reconfigured by the RL-based policy model in any use case and the computational resources at the disposal of the user (which could significantly reduce the necessary time, assuming potentially much larger cloud resources available to industry/Government stakeholders).

2.6 Training Process

“Both the MLP-based policy and the GCAPS-based policy are trained on an Intel Xeon Gold 6330 CPU (including 28 cores) with 512GB RAM and an NVIDIA A100 GPU. Note that this is expected to be a one/few-run offline investment for any given or existing network. Moreover, such (or even better) computing resources are readily available nowadays, making the training process a reasonable offline investment for training a real-time decision-support system (the policy models) for network reconfiguration. This solution strategy is particularly attractive considering that the real-time models are much faster than current baselines, as seen from the comparisons in section 2.12.”

Figure 4: Comparison of the performance of the different models with varying networks.

Review Comment #5:

The paper introduces a feature vector called the context, which includes variables such as energy supplied, voltage violation, and power flow through the edges. How critical are these contextual variables in the overall performance of the model?

Authors' Response:

We thank the reviewer for raising this question. The context vector encodes the state space variables that cannot be readily represented in the graph space, but are required to correctly compute the actions to be taken. These include the energy supplied, voltage violation, and the edge power flow. Since they represent a part of the state space (as explained in Section 2.3 under the subsection “State”), this information is **necessary** for computing the actions under the given MDP.

To provide some physical understanding, these context parameters can also be perceived as the feedback from the environment (e.g., error feedback in a control problem) that drives or constrains the actions that need to be taken. For instance, any control action adopted must not violate the safe and stable operational limits of the power distribution network. For a critical infrastructure such as the power network, voltage violations have serious ramifications. The goal in the steady state operation is to maintain the voltage in the network to prevent under-voltage and subsequent blackout. Similarly, the impact of the switching measured using energy supplied drives the control model to learn the optimal control policy and is therefore included in the context. The power flow through the edges is another useful information as it represents the functional status (on, off,

or outage) of the lines along with DN state. Some of these physical perspectives regarding the necessity of the “context” parameters (that serve as state variables) have now been discussed in the revised manuscript, as quoted below.

2.5 Learning Architecture (Context)

“The measurement of the impact of a control action on the distribution network performance serves as the context for training the model to embrace control policies that are both operationally feasible and safe. In the case of power networks, voltage violations can lead to severe consequences. The objective during steady-state operation is to uphold network voltage to prevent under-voltage and the ensuing blackout. Additionally, switching induces alterations in the network state, consequently causing a shift in the supplied energy. This impact is also considered as contextual information for the learning model. Similarly, the power flow through the branches which is representative of the DN state and line status (on, off, or outage) is encompassed within the context.”

Review Comment #6:

In Figure 3(a), it’s noteworthy that the performance of GCAPS declines at step 1.25, while MLP shows a rise in reward collection. Similarly, Figure 3(b) reveals that the network’s performance deteriorates after 800,000 steps. Given these observations, I wonder why the authors chose not to halt the training before reaching this point. Additionally, similar to the case of the 13-bus network, MLP’s performance eventually converges with that of GCAPS. This raises the question: if the number of training steps for MLP was increased, would it eventually outperform GCAPS?

Authors’ Response:

We thank the reviewer for raising this point. It should be noted that most RL algorithms optimize for a loss function where the loss includes policy gradient loss, value loss, and entropy loss. Therefore even though the reward has reached a maximum, it does not necessarily mean that the loss has converged. A trained network where the losses have not converged could lead to a very unstable policy model, such as producing unpredictably big changes in output action for a small change in the state space variables; this could happen the policy gradient loss has not converged. Hence, it is always safe to use the network weights where the loss has converged. However, in the updated training process, we used a squared exponential decreasing learning rate, with an initial high learning rate of $1e-5$ (which also addresses Comment 7 by Reviewer 1). With this new learning rate strategy, we do not encounter any unexpected deterioration in performance for both networks (GCAPS and MLP) for 13, 34, as well as for the 123 bus test networks. The training of the networks showcased a smoother convergence, as can be seen from Figures 5a, 5b, and 5c. The corresponding changes in the manuscript are quoted below:

4.3 Training Details

“Here we implement a squared exponential decreasing learning rate strategy $\rho_t = \rho_{\text{init}} \times e^{-(1-t)^2 \times D_R}$, where t represents the fraction of current step to the total number of steps for learning, ρ_t is the learning rate at t , ρ_{init} is the initial learning rate, and D_R is the decay rate. We used $\rho_{\text{init}} = 1e - 5$, and $D_R=3$. This strategy leads to smoother convergence and likely mitigates getting stuck in local minima. ”

Figure 5: **The training convergence plots for the policy models.** Our GCAPS-based GRL model is compared with the MLP model which does not utilize graph abstraction. (a) Convergence plot of GCAPS and MLP for the 13-bus network. (b) Convergence plot for the 34-bus network. (c) Convergence plot for the 123-bus network.

Review Comment #7:

The paper specifies a rather small learning step size, which raises concerns about the model potentially getting stuck in local minima during training. It might be beneficial for the authors to initially set a higher learning rate for the first 50 epochs and then reduce it by a factor of ten for subsequent phases, specifically the next two sets of 100 and 150 epochs. This suggestion contrasts with the paper’s current approach, which maintains a constant learning rate across all 100 epochs.

Authors’ Response:

We agree with this point and our response to the previous review comment already touches on this point. Moreover, note that, the high initial learning rate was found to significantly help improve the training performance of both GCAPS and MLP for the 13 and 34 bus networks by converging to a large average episodic reward. The corresponding change has been made in the training process description (Section 2.6) and training details (Section 4.3) of the revised manuscript, as quoted below:

2.6 Training Process

“For the 13-bus network, the average episodic reward for MLP converges to a slightly lower value than the peak value, while for GCAPS, the average episodic rewards are much higher compared to MLP, but couldn’t fully converge in 2 million steps. For 34-bus and 123-bus networks, GCAPS has a faster convergence compared to that of MLP.”

4.3 Training Details

“The training process is allocated a maximum of 36 hours, and the total number of steps is set to 2 million. For the 13-bus network, both GCAPS and MLP successfully completed the training with 2 million steps. However, for the 34-bus network, MLP could only be trained for 1.5 million steps within a 36-hour time frame, while the network for 123-bus systems could only be trained for 500,000 steps. To ensure a fair comparison, we utilize the trained weights of GCAPS and MLP at 1.5 million steps for the 34-bus network, and 500,000 steps for the 123-bus network. ”

Review Comment #8:

Given that the paper's primary contribution lies in the proposed algorithm, one would expect the authors to make the code available for replication and further scrutiny. Unfortunately, this opportunity has not been provided, neither for reviewers nor for the broader readership.

Authors' Response:

While it is not necessarily a common practice to make the codes available prior to the related work being fully accepted for publication, we have proactively prepared and made the codes to reproduce the results in this paper publicly available at:

<https://github.com/adamslab-ub/Real-Time-Outage-Management-Active-DNR-GRI>.

This information has also been included as a footnote in the updated manuscript on page 14 and page 30.

Authors' Response to Reviewer 2 Comments

We would like to express our gratitude to the reviewer for comprehensively reading our paper and providing valuable comments to help us improve the quality of our paper. Below, we address each comment in detail, aiming to enhance the overall quality of our work.

Review Comment #1

How do you determine or set the number and location of the switches in the smart grid? It has a significant impact on the performance of the approach and the computational burden. Do you optimize them? If Yes, How?

Authors' Response:

We thank the reviewer for these questions. The developed model was validated using modified IEEE test networks, incorporating sectionalizing and tie switches. The selection and placement of these switches were informed by insights gained from prior studies on power distribution networks.

We resort to leveraging existing studies for the placement of switches for the following reasons:

- The distribution test networks have been simulated using the OpenDSS software in this study. Therefore, modifying the standard 13, 34, and 123-bus networks necessitates a realistic and detailed description of the switches. While these networks are designed primarily for testing and research, standardized detailed information on switch properties, including phase connections, open/close states, line resistance, etc., is unavailable. Hence, we rely on well-established studies that provide coherent and meaningful component descriptions, having already validated the technical feasibility of circuits with the proposed modifications. This guarantees that simulating the network with varying switching controls does not yield erroneous results. Specifically, we have drawn upon the work of W.H. Kersting, as documented in [7], who conducted a comprehensive analysis of the 13-bus network, determining the appropriate types and locations of switches for this network. Furthermore, detailed switch properties for the 13-bus network found in [8] and [1] have been used in our study. Similarly, we used the switch information documented in [9] and [1] to describe the switches for the 34 and 123-bus networks in OpenDSS, respectively.
- Our model assumes that switches are already installed in the network, and therefore their relevant data is readily available as input for our decision-making tool. It is important to note that our focus does not encompass the optimization of switch locations and quantities. This is because one such task demands a planning study rooted in techno-economic analysis, a scope that extends beyond the objectives of our current work. However, as pointed out by the reviewer, the number of switches does indeed impact the computational burden on the model, analysis of which is evident from the comparison of results on 13/34/123 bus networks. The location of the switches, on the other hand, may or may not affect the generalizability of the learning process. However creation of multiple switch location scenarios, hence many different instances of the networks, is far from trivial since each of those have to be carefully crafted to be realistic, and is not within the scope of this work.
- In alignment with our study, which centers on autonomy decision-efficiency in outage management, we focus on remote-controlled switches. While the addition of more switches has the potential to reduce customer interruptions and outage duration in distribution networks, it also comes with substantial capital and operational costs. In [10], an optimal switch placement study was performed considering

these factors and it showed that the **optimal number of remote-controlled switches** falls within the range of 8 to 9 for the 37-node network while it is 15 to 22 for the 137-node network. Our research aligns with these findings, and we've taken these considerations into account while defining switches in the networks. We have used nine sectionalizing and tie switches for the 34-bus network and twenty two switches for the 123-bus network). Given that, switches are typically not deployed along all lines within the distribution network, our approach closely mirrors real-world conditions and constraints.

To clarify our settings in response to the above points, we have added the following descriptions to the revised manuscript.

2.7 Case Study on 13-bus Network: Scenario 1

“The quantity, positions, and specifications of the switches within the 13-bus test network are based on established studies that have previously validated the technical viability of these components within the circuit. Specifically, for the 13-bus network, we refer to the details presented in [1,7] to define the sectionalizing and tie switches. Our model assumes that switches are pre-installed in the network with their data available for our decision-making tool. However, optimizing switch locations and quantities falls within a planning study and requires a techno-economic analysis, which is beyond the scope of this paper. Our focus is on evaluating the model for enhancing operational resilience in power networks.”

2.9 Case Study on 34-bus Network: Scenario 1

“The details regarding the switches in the 34-bus network are adopted from [9], albeit presented in a different ordering of sectionalizing and tie switches here.”

2.11 Case Study on 123-bus Network

“The specifications of the switches are obtained from [1], albeit with a different arrangement in our implementation.”

2.12 Comparison of the Proposed Model with Baselines

“Although the computational complexity of the proposed model is contingent on the number of switches, the study in [10] found that the optimal number of remote-controlled line switches is 8 to 9 for a 37-node network and 15 to 22 for a 137-node network. Our research aligns with these findings, as we have considered nine sectionalizing and tie switches for the 34-bus network and twenty-two switches for the 123-bus network. This approach closely reflects real-world conditions and constraints, as switches are typically not deployed along all lines within the distribution network.”

Review Comment #2

Do you consider a penalty to show the forbidden states for the agent?

Authors' Response:

We thank the reviewer for bringing up this question. To address the possibility of the decision agent to take actions that leads to forbidden states during training, we have incorporated penalties into the reward. As the reward guides the learning process, this penalty discourages the agent from taking actions that result in invalid states of the network. Our reward formulation, detailed in Section 2.3, Subsection: Rewards (Eq.

1), introduces the term C_{viol} . This term is calculated following an action implementation and addresses the network infeasibility resulting from computed actions. Specifically, the energy supplied to the network is considered as a reward only if C_{viol} is zero, signifying successful power flow convergence. Challenges in achieving convergence would in this case indicate network issues, potentially arising from significant topological changes that render the network ill-conditioned or suffering from voltage deviations. Additionally, we have included a penalty term, V_{viol} , aimed at minimizing voltage violations throughout the network. This penalty term encourages the agent to take actions that keep the voltage levels within an acceptable range, ensuring that the network operates within safe and stable conditions.

In addition to avoiding forbidden states, we have incorporated a masking mechanism to prevent the selection of actions associated with the operation of outage switches. These switches (as explained in Section 2.3 under the Subsection: State), are excluded from the set of permissible actions. This mechanism ensures that the agent does not consider actions related to faulty switches or those used for isolating faults, thereby promoting compliant decision-making.

To clarify the use of constraints imposed via penalty and/or masking, we have updated the manuscript with the following modified text:

2.3 Formulating a Markov Decision Process over Graphs: 4. Reward

“To account for the possibility of the agent taking actions that could lead to forbidden states during its training, a penalty term is introduced into the reward. The reward, responsible for guiding the learning process, is augmented with this penalty to discourage the agent from pursuing actions that would result in invalid states. Specifically, we incorporate a term, denoted as C_{viol} , to address the power flow non-convergence issues. Such issues commonly arise during significant topological changes that render the network ill-conditioned, leading to an infeasible configuration, or when voltage levels deviate significantly from the normal operating range. Moreover, we include another penalty term, V_{viol} , aimed at minimizing voltage violations across the network. This term encourages the agent to take actions that maintain the voltage levels within an acceptable range, ensuring that the network operates in safe and stable conditions.”

2.3 Formulating a Markov Decision Process over Graphs: 1. State

“The variable \mathcal{O} in the state space reflects the outage scenario, i.e., the multi-line failures in the network, including switch outages. The inoperability of the outage switches is addressed by using a masking mechanism that assures the suppression of the corresponding switching action, represented by the state variable μ .”

Review Comment #3

Is this method centralized or decentralized? considering one single agent to control all switches together is not preferable. Because, in this case, you will need a strong communication system its own resilience will be a critical issue, which it seems it has been ignored in this paper.

Authors' Response:

We thank the reviewer for these insightful questions. The outage management tool, in this study, has been employed on test feeders featuring a **single substation** supplying power to loads at the primary level of distribution transformers while also incorporating integrated distributed energy resources into the network. In a distribution network, the **distribution system operator (DSO)** is responsible for regulating power balance

and controlling the resources to ensure safe and stable operation. Here, the role of the DSO is assumed by a centralized autonomous agent capable of reconfiguration decision-making. Additionally, note that the **switches are within the distribution network ownership. Consequently, a centralized approach for outage management is deemed practical**, as the ultimate decision-making authority rests with the DSO or a designated substation agent.

Further reasons for adopting the centralized approach based on a single agent are as follows:

- Despite various stakeholders in power distribution, the DSO, or a substation agent, ultimately oversees the process. In contrast, tasks such as demand response during extreme events and control for power network protection are typically addressed using a decentralized architecture, given the presence of multiple decision-making entities. For instance, in demand response, the prosumers participate in decision-making alongside the DSO, while in the protection system, relays and circuit breakers dispersed throughout the network use local information to determine control actions. However, outage management presents a unique challenge, and our **problem formulation is tailored** specifically to this task. Therefore, rather than defaulting to a decentralized approach, which may not align with the hierarchy of real-world networks, we consciously adhere to a centralized approach.
- While decentralized Multi-Agent Systems (MAS) prove valuable in addressing complex network planning problems, their application is suitable for scenarios where distinct tasks are delegated to various organizations or entities, called agents [11], or observation is expected to be partial. These agents store and process local private information to pursue individual goals within the framework of an overarching objective. In contrast, as discussed earlier, the central regulatory body (DSO) exerts a monopoly over system control, operating all switches to optimize the overall system energy supply while using the global state variables. Decentralized control systems, on the other hand, base decision-making on local variables. This rationale extends to the protection system and sequential load pick-up in a de-energized network during black start, where the state or measurements of adjacent loads (nodes) influence decision control. Conversely, **outage management using reconfiguration necessitates information from the wider area system**. Additionally, one of our future endeavors involves applying this approach to practical distribution systems that may not be fully observable, implying limited sensor placement throughout the network. Consequently, information in the local neighborhood of a switch may be unavailable and the approach will be to use the limited wide-area (global) measurements for resilient decision support.
- The motivation behind using the GCAPS method as the policy model was to achieve a better infusion of global information alongside the local node properties. This choice resulted in the model attaining near-global optimal performance. In contrast, **MAS-based system** control, while computationally efficient, may **struggle to consistently achieve the optimal results** [11]. Considering the critical nature of the distribution network, the objective is to swiftly obtain near-optimal stable solutions. The demonstrated timeliness of decision-making in the GCAPS model reinforces our approach of treating the substation as an agent utilizing global information within its zone for near-optimal decision-making.
- Additionally, it is important to note that **the primary focus of this study is not on designing an MAS architecture**, as seen in other works [12–14]. In those works, the authors proposed an MAS control system design for performing service restoration, to improve reliability and reduce single-point failures. This further requires optimizing the communication network including the number of communication channels. Contrary to this, our objective in this work is not to define the control flow within the smart grid. We operate under the assumption that the existing control architecture, with a DSO (in this case, an autonomous agent), is already in place. While our study acknowledges the evolving nature of

control architectures in smart grids, with a potential shift towards distributed control, clear standards in this domain are currently lacking. Consequently, we opt to work within the existing framework, leveraging novel Graph Reinforcement Learning (GRL) as a key element for automatic and rapid response. However, we acknowledge that if a shift to a decentralized control structure occurs in the future, our problem formulation should be adapted accordingly. The GRL architecture that we use here has already been used by us in decentralized settings in other MAS type applications such multi-robot task allocation (incl. under communication uncertainty) [15–17]; and hence there isn't a fundamental challenge to reformulating and solving our problem for a decentralized setting with further extensions to the current GL architecture.

We thank the reviewer's emphasis on the necessity of a robust communication system for effective outage mitigation. While the communication infrastructure in a decentralized MAS may be streamlined, it does not inherently address the challenge of communication failures especially in the case of natural or anthropogenic hazards that cause the power grid failures. It is **crucial to recognize and evaluate the impact of communication breakdowns on resolving power network outages**, as discussed in [18]. This requires delving into the coupling of the distribution grid with the communication network, detailed cyber-physical modeling, evaluating their interactions, and the formulation of a communication recovery strategy. Models that do account for these features at a realistic level of complexity remains lacking. For the scope of this paper, our focus is specifically on modeling and recovering the power network, with an assumption of the communication network's functional status. Tackling both networks within a unified framework poses a challenge. Consequently, we view this as a potential extension of our work, wherein outage management for the interconnected power and communication networks will be modeled using GRL applied to multi-layered graphs.

To clarify the utility of the proposed approach and the control architecture assumed, we have added the following portions in the revised manuscript:

2.3 Formulating a Markov Decision Process over Graphs

“The outage management tool is applied to power distribution networks where the distribution system operator (DSO) or substation agents are responsible for regulating the power balance and controlling the resources to ensure safe and stable operation. In this study, the test feeders under consideration feature a single substation supplying power to loads while integrating distributed energy resources. Consequently, we adopt a centralized approach for outage management, treating the DSO or substation agent as an autonomous decision-making entity.

The formulation of our approach for outage management is tailored to align with the control architecture found in real-world distribution networks, instead of defaulting to a decentralized approach. Besides this, a multi-agent system (MAS) based approach may prove unsuitable for reconfiguration which relies on wide-area measurements, especially in networks where observability is limited, and local information is constrained. Additionally, the MAS while computationally efficient, encounters challenges in consistently achieving the optimal results [11]. On the other hand, the developed GCAPS with centralized control can achieve near-optimal results by integrating global (wide-area) and local properties into the learning model. It is crucial to highlight that the primary focus of this study does not revolve around designing an MAS architecture, as seen in other works [12, 13]. Our objective is not to prescribe the control flow within the smart grid, and we operate under the assumption that the existing control architecture, with a DSO (in this case, an autonomous agent), is already established. While acknowledging the evolving nature of control architectures in smart grids, with a potential shift towards distributed control, it is essential to note the current lack of clear standards in this domain.”

3 Discussion

“It is also crucial to model and evaluate the impact of communication breakdowns on resolving power network outages, since those can be an associated artifact attributed to the natural or anthropogenic hazard that caused the power grid breakdown. This, however, necessitates intricate coupled cyber-physical modeling of the communication network, and formulation of communication recovery as in [18]. Addressing the modeling and control of coupled communication and power networks as a unified effort poses significant challenges. A potential extension of our work involves modeling the interconnected power and communication networks as multi-layered graphs and evaluating the impact of communication failure on power network recovery.”

Review Comment #4

The test (13-bus) system is too small. Using this test case, it is not possible to show the time-efficiency of the methodology.

Authors’ Response:

We thank the reviewer for this comment. While it is true that the 13-bus network is relatively small, it represents the type of network we intend our model to handle. This network although small, is subjected to high loading conditions for a 4.16 kV feeder, encompassing unbalanced spot and distributed loads. Additionally, it also consists of both overhead and underground lines with varying phases. As this network possesses all the characteristic features of a typical distribution network, we deemed it suitable for a small-scale implementation of our model. Additionally, we aimed to use this network to illustrate the behavior preceding and following control actions during outages. Given its compact size, readers will find it easier to comprehend the impact of outages and the subsequent actions taken, making it more visually intuitive. As per both reviewers’ suggestions, we have also added a case study on the 123-bus network in addition to the 34-bus network in this paper. **Our response to comment #1 from Reviewer 1 provides more details of this significant change, including quotations of the comprehensive additions made w.r.t. to the new 123 bus case study in the revised manuscript.**

Review Comment #5

Does this method have the possibility to be applied to large-scale systems?

Authors’ Response:

We thank the reviewer for raising this question. As per the reviewer’s comment, we have also evaluated the outage management model on the 123-bus distribution network. While distribution networks are typically large-scale systems, our model has the potential for extension to accommodate such networks. As demonstrated in this paper, the rapid decision-making capability, as opposed to traditional methods, is a key strength of our model. Traditional methods experience an increase in response time as the network size grows, whereas our model demonstrates real-time decision making capability even with larger networks. However, it is important to note that dealing with larger networks presents challenges during the training phase, as it requires advanced computational resources to adequately train the learning over the graphs model. Given the improved operational resilience and distribution intelligence offered by the model, allocating resources for its training is reasonable. This limitation is only encountered in the offline phase, while the online phase is expected to perform well. To further illustrate the scalability of the proposed model, we have also included a section describing the case study on the 123-bus network in the revised manuscript. The corresponding

changes have been made in Section 2.11 (quoted in the response to comment 1 of Reviewer 1), and Section 2.12 (quoted in the response to comment 1 of Reviewer 1), and Section 3 (quoted below) of the revised manuscript:

3 Discussion

“As demonstrated in this paper, the rapid decision-making capability in contrast to traditional methods, is a key strength of our model. Unlike conventional approaches, our model demonstrates real-time response times to increasing network size, making it well-suited for online deployment on large distribution networks. However, it is important to note that dealing with larger networks presents challenges during the training phase, demanding advanced computational resources to adequately train the learning over the graphs model. This limitation is encountered during the offline phase and can be resolved by allocating adequate resources for training, considering the benefit of operational resilience.”

Review Comment #6

Working with one smart grid to control it needs communications with the other neighboring smart grids. Any ideas for it? Please elaborate.

Authors' Response:

The smart grid encompasses various elements of the electric grid, ranging from power generation to transmission and distribution to end-users. This transformation of the traditional grid to the smart grid is characterized by the integration of digital technology, facilitating enhanced communication, advanced monitoring, and intelligent control. Notably, this transition advocates a **bottom-up approach** [19]. Therefore, the lower-level component of the smart grid, specifically the **distribution feeders, interacts and communicates with each other at the upstream transmission level. This paper delves into the specifics of the lower-level component of the smart grid, focusing on the distribution network.** The smart grid itself, on the other hand, typically **operates as an independent entity governed by an independent system operator (ISO).** Conversely, the communication and coordination between multiple smart grids are classified as intergrid operations. However, the intergrid operation is dependent on different factors such as geographical location, regulatory frameworks, and the level of coordination between the different ISOs [20]. This becomes particularly challenging when the smart grids are managed by separate ISOs as they have different operational protocols, communication systems, and regulatory jurisdictions. It is important to note that the standardization of smart grid communication infrastructure and interoperability is still evolving [21]. This ongoing development is a crucial aspect to consider in the context of interconnected smart grids (at the transmission level).

Distribution intelligence is an integral component of the smart grid [19]. Within the smart grid framework, distribution networks utilize advanced metering and communication technologies to offer much-needed situational awareness to distribution system operators (DSO) at the lower level. The control actions adopted by the intelligent control mechanisms and **the concurrent state of individual distribution networks interconnected by the transmission system are available in real time to the smart grid operating agents** [20]. This enables them to attain a comprehensive understanding of the overall grid status. Concerning the state of the transmission system, each distribution feeder is considered lumped into an equivalent load at the substation buses. Furthermore, the distribution system incorporates local energy resources capable of sustaining the network during outages, even in isolation from the main grid (off-grid mode considered in this paper). Therefore, the **operation and control of distribution networks, even during outages, function**

autonomously, relying on the status of the distribution network components (including the substation). The transmission network on the other hand requires the consideration of the interconnected operation of the distribution feeders, which is out of the scope of this work. Additionally, **intergrid operation is rarely employed** during extreme events due to the risk of failures propagating from one independent system to the other.

The following description has been added to the manuscript to elaborate on this comment:

1 Introduction

“The transformation of the grid into a smart grid is driven by a bottom-up approach [19] with distribution feeders interacting at the transmission level. This paper specifically explores the intricacies of the lower-level component of the smart grid - the distribution network. The smart grid typically operates as an independent entity governed by an independent system operator (ISO). Intergrid operations are challenging due to differing protocols, communication systems, and regulatory jurisdictions among ISOs. Additionally, exploring new frontiers in smart grid operation is constrained by the ongoing development of communication infrastructure standardization and interoperability. The operation and control of the distribution networks within the smart grid are mostly autonomous with its aggregated impact visible on the transmission level [20]. However, inter-grid operations are seldom employed during extreme events, driven by concerns about potential cascading failures between independent entities.”

References

- [1] M. Quintero-Duran, J.E. Candelo, J. Soto-Ortiz, A modified backward/forward sweep-based method for reconfiguration of unbalanced distribution networks. *International Journal of Electrical & Computer Engineering* (2088-8708) **9**(1) (2019)
- [2] A. Arif, Z. Wang, Networked microgrids for service restoration in resilient distribution systems. *IET Generation, Transmission & Distribution* **11**(14), 3612–3619 (2017)
- [3] A.J.M. Muzahid, S.F. Kamarulzaman, M.A. Rahman, in *2021 International Conference on Software Engineering & Computer Systems and 4th International Conference on Computational Science and Information Management (ICSECS-ICOCSIM)* (IEEE, 2021), pp. 200–205
- [4] A.L. Bayona Latorre, Comparative study of SAC and PPO in multi-agent reinforcement learning using unity ml-agents (2023)
- [5] A. Raffin, A. Hill, A. Gleave, A. Kanervisto, M. Ernestus, N. Dormann, Stable-baselines3: Reliable reinforcement learning implementations. *Journal of Machine Learning Research* **22**(268), 1–8 (2021). URL <http://jmlr.org/papers/v22/20-1364.html>
- [6] H. Sekhavatmanesh, R. Cherkaoui, A novel decomposition solution approach for the restoration problem in distribution networks. *IEEE Transactions on Power Systems* **35**(5), 3810–3824 (2020)
- [7] W. Kersting, in *2014 IEEE Rural Electric Power Conference (REPC)* (IEEE, 2014), pp. B3–1
- [8] S. Chakrabarti, G. Ledwich, A. Ghosh, in *2009 International Conference on Power Systems* (IEEE, 2009), pp. 1–6
- [9] P. Gangwar, S.N. Singh, S. Chakrabarti, Network reconfiguration for the DG-integrated unbalanced distribution system. *IET Generation, Transmission & Distribution* **13**(17), 3896–3909 (2019)
- [10] M. Jooshaki, S. Karimi-Arpanahi, M. Lehtonen, R.J. Millar, M. Fotuhi-Firuzabad, An MILP model for optimal placement of sectionalizing switches and tie lines in distribution networks with complex topologies. *IEEE transactions on smart grid* **12**(6), 4740–4751 (2021)
- [11] A. Sujil, J. Verma, R. Kumar, Multi agent system: concepts, platforms and applications in power systems. *Artificial Intelligence Review* **49**, 153–182 (2018)
- [12] A. Elmitwally, M. Elsaid, M. Elgamal, Z. Chen, A fuzzy-multiagent service restoration scheme for distribution system with distributed generation. *IEEE Transactions on Sustainable Energy* **6**(3), 810–821 (2015)
- [13] G. Rohbogner, S. Fey, P. Benoit, C. Wittwer, A. Christ, Design of a multiagent-based voltage control system in peer-to-peer networks for smart grids. *Energy Technology* **2**(1), 107–120 (2014)
- [14] A. Elmitwally, M. Elsaid, M. Elgamal, Z. Chen, A fuzzy-multiagent self-healing scheme for a distribution system with distributed generations. *IEEE transactions on power systems* **30**(5), 2612–2622 (2014)
- [15] S. Paul, P. Ghassemi, S. Chowdhury, in *2022 International Conference on Robotics and Automation (ICRA)* (IEEE, 2022), pp. 8815–8822
- [16] S. Paul, S. Chowdhury, in *AIAA AVIATION 2022 Forum* (2022), p. 3911

- [17] S. Paul, W. Li, B. Smyth, Y. Chen, Y. Gel, S. Chowdhury, Efficient planning of multi-robot collective transport using graph reinforcement learning with higher order topological abstraction. arXiv preprint arXiv:2303.08933 (2023)
- [18] X. Wang, Q. Kang, X. Wei, L. Guo, Z. Liang, Resilience assessment and recovery of distribution network considering the influence of communication network. *International Journal of Electrical Power & Energy Systems* **152**, 109280 (2023)
- [19] Distribution intelligence. https://www.smartgrid.gov/the_smart_grid/distribution_intelligence.html
- [20] Z. Fan, Y. Mao, T. Horger, in *IEEE PES T&D 2010* (IEEE, 2010), pp. 1–8
- [21] A. Gopstein, C. Nguyen, C. O’Fallon, N. Hastings, D. Wollman, et al., *NIST framework and roadmap for smart grid interoperability standards, release 4.0* (Department of Commerce. National Institute of Standards and Technology ..., 2021)

REVIEWER COMMENTS

Reviewer #2 (Remarks to the Author):

Dear Authors,

Thank you for your significant work. My comments are addressed adequately.

Reviewer #3 (Remarks to the Author):

1. In Equation 1, the authors are giving a reward of zero when the voltage of the system does not converge due to the action being infeasible and they are giving a negative reward for violating system constraint. This kind of reward formulation would encourage the agent to prioritize taking infeasible actions over actions that are feasible but cause voltage violation. How do the authors justify the reward function? Have the authors checked the feasibility of the solutions they have obtained?

2. In reinforcement learning, the agent does some trial and error at the beginning until it learns the policy. Based on the reward graphs provided, it seems that the agent reward is always increasing as if the agent always knows which action to take to increase the reward. The authors should elaborate on this.

3. The authors should also include more of the recent work in reinforcement learning in distribution networks in their literature review section.

Reviewer #3 (Remarks on code availability):

I did not run the code but the content of the code is OK.

Authors Response to Reviews of

Real-Time Outage Management in Active Distribution Networks Using Reinforcement Learning over Graphs

Roshni Anna Jacob, Steve Paul, Souma Chowdhury, Yulia R.Gel, and Jie Zhang

Authors' Response to Review 3 Comments

Review Comment #1:

In Equation 1, the authors are giving a reward of zero when the voltage of the system does not converge due to the action being infeasible and they are giving a negative reward for violating system constraint. This kind of reward formulation would encourage the agent to prioritize taking infeasible actions over actions that are feasible but cause voltage violation. How do the authors justify the reward function? Have the authors checked the feasibility of the solutions they have obtained?

Authors' Response:

We thank the reviewer for raising this question. The term associated with non-convergence C_{viol} indicates the reliability of the state acquired from the OpenDSS solver performing the power flow analysis. Since the state of the network may not accurately or conclusively represent the impact of the switching actions on the DN when the power flow simulation itself exhibits non-convergence, we have assigned it a neutral reward. However, the infeasibility in DN operation corresponds to significant voltage violations and thus imminent network collapse, which is therefore considered as a penalty when formulating the reward. This approach guides the agent towards attaining non-zero positive rewards. These choices are further explained below.

Following the implementation of any control mechanism on the power distribution network, we evaluate the network state using the OpenDSS simulation tool which performs power flow analysis. The OpenDSS solver employs a fixed-point iterative numerical approach to solve the power flow, utilizing a preset threshold (e.g., 0.0001) on nodal voltage magnitude to determine the convergence of power flow. The outage of specific elements in the power distribution network can lead to network topology comprising multiple isolated segments. While reconfiguration may mitigate this issue to some extent by establishing alternative routes, some sections may remain isolated, housing several active components such as transformers, regulators, loads, and generators, all described within the DSS circuit. The inability of the OpenDSS solver to determine the state (or control) variables associated with these components, and the absence of a strong slack bus (as the substation) may result in the network being ill-conditioned. In such scenarios, we have observed that some sections of the network may be able to achieve the desired nodal power balance while others may not. However, the DSS circuit is always solved as a whole. Therefore, observed non-convergence indicates that the DSS runs the power flow up to a few thousand iterations without achieving nodal power balance, without a mismatch of less than 0.0001 for the entire network (sectionalized or otherwise). In such cases, the observation fails to accurately reflect the actual network state. Consequently, it cannot be concluded if the switching control is indeed infeasible. Since these actions could not be attributed a positive or negative value, we assigned a neutral value (0). We acquire the convergence flag from the OpenDSS tool to determine if the network state is reliable. This is the rationale for C_{viol} in the reward function. Here $C_{viol} = 0$ indicates that the power flow has converged (mismatch at all nodes in the network <0.0001) and a non-zero value

indicates non-convergence. Consequently, we assign a non-zero value to the reward only when the power flow converges, as seen in Eq. (1).

On the other hand, the network infeasibility stems from the voltage violations themselves. Exorbitantly large voltage violations render the network operation infeasible and is assigned a corresponding negative value. Such operating conditions are indicative of imminent network collapse. As observed in Eqs. (2), (3), and (4), the voltage violation is calculated by quantifying the deviation of node voltages from the desirable or safe operating limits. Smaller deviations indicate superior network performance and also the sustainable operation of the network. The optimal policy is determined by maximizing the reward. Therefore, according to our objectives of maximizing the energy supplied and minimizing the voltage violations, we formulate the reward as presented in the manuscript. Therefore, the policy is guided towards acquiring non-zero positive rewards through the learning process. From the rewards plot in Fig. 3, we see that the general trend of the model is to take actions that maximize the energy supplied and minimize voltage violations. The agent has also been observed to encourage this behavior in the solutions we examined.

Any potential ambiguity in the textual explanation provided in this regard in the earlier version of the manuscript has now been revised as follows:

2.3 Formulating a Markov Decision Process over Graphs

“Reward (\mathcal{R}): The reward reflects the goal of improving resilience in the DN by maximizing the energy supplied E_{supp} while minimizing violations of voltage constraints. To account for the network being ill-conditioned with specific outage conditions and switching actions, a term C_{viol} is introduced into the reward. The DN, subject to topology changes due to outages and switching actions, may consist of multiple independent sections (network components), each housing various active components (transformers, regulators, generators, loads, etc.) with corresponding state variables. In some scenarios, the isolation of these components from a robust slack (substation) renders the network ill-conditioned, resulting in challenges in achieving nodal power balance within a preset tolerance of mismatch. This lack of balance in certain sections of the DN leads to non-convergence of power flow, identifiable through flags in the solver. This issue is attributed a zero value with the actual impact of switching on the network state being indeterminate given that the solver fails to accurately reflect the network behavior with switching. On the other hand, the network operation with large voltage violations is infeasible as it leads to immediate network collapse. To discourage the agent from pursuing actions that result in actions leading to invalid states, the reward is augmented with a penalty term, V_{viol} . The goal here is to maintain the voltage levels within an acceptable range, ensuring that the network operation is sustainable.”

Review Comment #2:

In reinforcement learning, the agent does some trial and error at the beginning until it learns the policy. Based on the reward graphs provided, it seems that the agent reward is always increasing as if the agent always knows which action to take to increase the reward. The authors should elaborate on this.

Authors’ Response:

It should be noted that the plots in Fig. 3 represent the average episodic reward and not the reward for every step or every batch. The average episodic reward is computed and reported in the training history after the policy updates made over an entire rollout. Each rollout operation consists of N_{steps} , which is divided or grouped into N_{batch} number of batches, with each batch being trained for 100 epochs. Even though the reward

for every step or even every batch during the rollout operation need not always increase and can fluctuate, it is quite reasonable for the average episodic reward to increase (without significant oscillations), provided the learning rate is not too high.

To further elaborate, in this work, we employed Proximal Policy Optimization (PPO) [1] for the training, where the policy is updated after every rollout operation (consisting of N_{steps}). During the policy update, the weights of the policy network are adjusted to minimize the loss function, consisting of value loss, entropy loss, and policy advantage loss. PPO takes a constrained weight update that usually guarantees an average episodic reward increase, even if it is by a small margin. We also provide an entropy coefficient of 0.1, which enforces an explorative behavior for policy learning. For example, the training history plot from the original paper on PPO [1] shows consistently increasing rewards, which is similar to our plots (Fig. 3).

We have also added the following text in Section 2.6 for further clarity in this regard.

2.6 Training Process

“Figure 3 shows the training history in terms of the average episodic reward after each rollout, while training GCAPS and MLP for 13, 34, and 123 bus systems. The average episodic reward is computed as the average of the episodic rewards for all the episodes in each rollout operation.”

Review Comment #3:

The authors should also include more of the recent work in reinforcement learning in distribution networks in their literature review section.

Authors' Response:

We thank the reviewer for this suggestion.

We have now added recent works pertaining to reinforcement learning in power distribution networks to the literature review section (Section 1) in the revised version of the manuscript. The added portions are quoted below.

1 Introduction

“Deep RL is being increasingly employed for voltage control in active DNs in recent literature. In [2], the DER inverters and static VAR compensators were controlled to achieve the desired voltage levels in the network using a combination of graph-based network representation learning, surrogate model of power flow, and soft actor-critic algorithm. In [3], the distributed energy storage devices have been treated as agents, and a multi-agent deep RL was utilized for voltage regulation with the capability to respond to topology changes as well. In another study [4], multi-agent deep RL was applied to perform optimal scheduling of various DERs, energy storage systems, and flexible loads within the network. In this context, the inverters associated with DERs and energy storage can be considered as individual agents. The role of such devices in voltage regulation aligns with the distributed nature of their control mechanism. Conversely, outage management using reconfiguration and load control relies on wide-area measurements at the control center to facilitate switching operations.”

“In another work [5], a deep Q-learning-based RL approach was employed to dynamically form microgrids in response to outages. However, as discussed previously, this method necessitates the compilation of all radial feasible structures before the learning process and does not encompass both forms of reconfiguration. Similarly, [6] utilized a Q-learning-based strategy for

reconfiguration and load shedding. Lastly, in addition to load and switch control, deep RL could also be used for optimal dispatch of DERs in islanded mode as demonstrated in [7].”

References

- [1] J. Schulman, F. Wolski, P. Dhariwal, A. Radford, O. Klimov. Proximal policy optimization algorithms (2017)
- [2] D. Cao, J. Zhao, J. Hu, Y. Pei, Q. Huang, Z. Chen, W. Hu, Physics-informed graphical representation-enabled deep reinforcement learning for robust distribution system voltage control. *IEEE Transactions on Smart Grid* (2023)
- [3] Y. Xiang, Y. Lu, J. Liu, Deep reinforcement learning based topology-aware voltage regulation of distribution networks with distributed energy storage. *Applied Energy* **332**, 120510 (2023)
- [4] Y. Lu, Y. Xiang, Y. Huang, B. Yu, L. Weng, J. Liu, Deep reinforcement learning based optimal scheduling of active distribution system considering distributed generation, energy storage and flexible load. *Energy* **271**, 127087 (2023)
- [5] M.A. Igder, X. Liang, Service restoration using deep reinforcement learning and dynamic microgrid formation in distribution networks. *IEEE Transactions on Industry Applications* (2023)
- [6] L.R. Ferreira, A.R. Aoki, G. Lambert-Torres, A reinforcement learning approach to solve service restoration and load management simultaneously for distribution networks. *IEEE Access* **7**, 145978–145987 (2019)
- [7] Y. Du, D. Wu, Deep reinforcement learning from demonstrations to assist service restoration in islanded microgrids. *IEEE Transactions on Sustainable Energy* **13**(2), 1062–1072 (2022)

REVIEWERS' COMMENTS

Reviewer #3 (Remarks to the Author):

1) When the voltage violation happens, the action is infeasible. Similarly, if you are isolating a part of the system, this is still an infeasible action as distribution networks should remain radial at all times.

2) PPO does indeed employ a constrained optimization approach to update the policy network. This constraint helps to ensure that the policy does not deviate too far from the previous policy during updates, which can lead to more stable training and prevent large policy changes that might harm performance. However, while this constraint does aim to improve the policy iteratively, it doesn't strictly guarantee that the average episodic reward will always increase. The guarantee of reward increase, even by a small margin, is not inherent to the PPO algorithm itself.

Authors Response to Reviews of

Real-Time Outage Management in Active Distribution Networks Using Reinforcement Learning over Graphs

Roshni Anna Jacob, Steve Paul, Souma Chowdhury, Yulia R.Gel, and Jie Zhang

Authors' Response to Review 3 Comments

Review Comment #1:

When the voltage violation happens, the action is infeasible. Similarly, if you are isolating a part of the system, this is still an infeasible action as distribution networks should remain radial at all times.

Authors' Response:

We appreciate the reviewer for their insightful comments and engaging discussion. In our reward formulation, we address voltage violations by considering them as penalties. Traditionally, optimization problems such as those solved using the mixed integer second-order conic programming (MISOCP) framework—used as a baseline in our research—treat voltage violations as hard constraints. However, in reinforcement learning (RL), enforcing such constraints as hard limits i.e., eliminating solutions that violate them, is not practical. Instead, the approach adopted in RL is to penalize the agent for actions that lead to infeasibility.

While undervoltage is undesirable during normal operation, maintaining voltages within defined thresholds can become challenging during extreme events, such as when central generating stations drop offline and localized generation becomes predominant. In such circumstances, it may be prudent to leverage existing resources to sustain network supply and deploy additional resources to fortify the network. The RL model we present serves as a resilience-oriented decision-making tool, offering near-optimal actions. Implementing these actions facilitates swift and efficient emergency response. To mitigate subsequent undervoltages, deploying mobile energy resources in affected areas or resorting to further load shedding may be necessary based on the network state.

Power distribution networks typically operate in a radial configuration, where all nodes are connected to the source bus, usually a substation. However, during outages, while performing switching control, it may be necessary to operate the distribution network by isolating small sections to address emergency conditions as discussed in [1, 2]. In these scenarios, certain sections of the network can be isolated while still maintaining power supply for the rest of the network, depending on the available energy resources, to ensure load-generation balance. This approach is part of reconfiguration and islanding strategies, which need to be implemented optimally. The objective function is designed to maximize the power supply within the network while minimizing voltage violations, thereby effectively addressing these goals.

Review Comment #2:

PPO does indeed employ a constrained optimization approach to update the policy network. This constraint helps to ensure that the policy does not deviate too far from the previous policy during updates, which can lead to more stable training and prevent large policy changes that might harm

performance. However, while this constraint does aim to improve the policy iteratively, it doesn't strictly guarantee that the average episodic reward will always increase. The guarantee of reward increase, even by a small margin, is not inherent to the PPO algorithm itself.

Authors' Response:

We agree that during training the average episodic reward does not always increase over consecutive iterations. This can also be seen in all three training history plots in Figure 3 of the manuscript. The paper does not claim any strict monotonic increase in average episodic reward. Even though, overall, the reward is expected to increase across a finite number of iterations or epochs (when training is considered to be progressive), as is the case in our results, minor oscillations are a common artifact of PPO and other policy gradient approaches; this is due to the randomness of the batch compositions and stochastic gradient search used. As we are employing a standard PPO algorithm (from the stable-baselines3 platform), this is just as expected.

References

- [1] C. Chen, J. Wang, F. Qiu, D. Zhao, Resilient distribution system by microgrids formation after natural disasters. *IEEE Transactions on smart grid* **7**(2), 958–966 (2015)
- [2] L. Che, M. Shahidehpour, Adaptive formation of microgrids with mobile emergency resources for critical service restoration in extreme conditions. *IEEE Transactions on Power Systems* **34**(1), 742–753 (2018)